# Prosthetic forefoot and heel stiffness across consecutive foot stiffness categories and sizes

**Anne T. Turner**[1,2☯], **Elizabeth G. Halsne**[1,3☯], **Joshua M. Caputo**[4], **Carl S. Curran**[4], **Andrew H. Hansen**[5,6], **Brian J. Hafner**[3], **David C. Morgenroth**[1,3☯]*

**1** VA RR&D Center for Limb Loss and Mobility (CLiMB), VA Puget Sound Health Care System, Seattle, Washington, United States of America, **2** Elson S. Floyd College of Medicine, Washington State University, Spokane, Washington, United States of America, **3** Department of Rehabilitation Medicine, University of Washington, Seattle, Washington, United States of America, **4** Human Motion Technologies LLC d/b/a Humotech, Pittsburgh, Pennsylvania, United States of America, **5** Minneapolis VA Health Care System, Minneapolis, Minnesota, United States of America, **6** University of Minnesota, Minneapolis, Minnesota, United States of America

☯ These authors contributed equally to this work.
* dmorgen@uw.edu

**Data Availability Statement:** All relevant data are within the paper and its Supporting Information files.

## Abstract

Prosthetic foot stiffness plays a key role in the functional mobility of lower limb prosthesis users. However, limited objective data exists to guide selection of the optimal prosthetic foot stiffness category for a given individual. Clinicians often must rely solely on manufacturer recommendations, which are typically based on the intended user's weight and general activity level. Availability of comparable forefoot and heel stiffness data would allow for a better understanding of differences between different commercial prosthetic feet, and also between feet of different stiffness categories and foot sizes. Therefore, this study compared forefoot and heel linear stiffness properties across manufacturer-designated stiffness categories and foot sizes. Mechanical testing was completed for five types of commercial prosthetic feet across a range of stiffness categories and three foot-sizes. Data were collected for 56 prosthetic feet, in total. Testing at two discrete angles was conducted to isolate loading of the heel and forefoot components, respectively. Each prosthetic foot was loaded for six cycles while force and displacement data were collected. Forefoot and heel measured stiffness were both significantly associated with stiffness category ($p = .001$). There was no evidence that the relationships between stiffness category and measured stiffness differed by foot size (stiffness category by size interaction $p = .80$). However, there were inconsistencies between the expected and measured stiffness changes across stiffness categories (i.e., magnitude of stiffness changes varied substantially between consecutive stiffness categories of the same feet). While statistical results support that, on average, measured stiffness is positively correlated with stiffness category, force-displacement data suggest substantial variation in measured stiffness across consecutive categories. Published objective mechanical property data for commercial prosthetic feet would likely therefore be helpful to clinicians during prescription.

**Funding:** This research is a project of the Seattle Institute for Biomedical and Clinical Research (https://www.sibcr.org/) supported by the Office of the Assistant Secretary of Defense for Health Affairs, through the Orthotics and Prosthetics Outcomes Research Program (https://cdmrp.army.mil/oporp/default) under Award No. W81XWH-16-1-0569 (PI: D.C.M.). Opinions, interpretations, conclusions and recommendations are those of the authors and are not necessarily endorsed by the Department of Defense. The U.S. Army Medical Research Acquisition Activity, 820 Chandler Street, Fort Detrick MD 21702-5014 is the awarding and administering acquisition office. This material is the result of work supported with resources and the use of facilities at the VA Puget Sound Health Care System, Seattle, Washington (https://www.pugetsound.va.gov/). The contents do not represent the views of the United States Department of Veterans Affairs or the United States Government. This study was not funded or sponsored by any commercial companies. Co-authors J.M.C. and C.S.C. are employed by Human Motion Technologies LLC d/b/a Humotech at no cost to the study. Humotech does not have any competing interests for the data presented in this manuscript. The study sponsor (the Office of the Assistant Secretary of Defense for Health Affairs, through the Orthotics and Prosthetics Outcomes Research Program under Award No. W81XWH-16-1-0569), provided support in the form of salaries for authors [D.C.M., E.G.H., A.T.T., B.J.H]. However, the funder did not have any role in the study design, data collection and analysis, decision to publish, or preparation of the manuscript. The specific roles of these authors are articulated in the 'author contributions' section.

**Competing interests:** Co-authors J.M.C. and C.S.C. are employed by Human Motion Technologies LLC d/b/a Humotech, but Humotech was not a funder for this study and the company has no competing interests that would be relevant to the work done on this manuscript. Therefore, this does not alter our adherence to PLOS ONE policies on sharing data and materials. No authors on this manuscript have any competing interests.

# Introduction

Prosthetic feet are an essential component of lower limb prostheses, and are intended to restore mobility to people with lower limb amputations (LLA).[1, 2] There are a vast number of commercially-available prosthetic feet with a wide range of designs and mechanical properties which are potentially suited to particular mobility levels or specific mobility activities.[3–5] These design variations correspond with different stiffness behaviors of the prosthetic forefoot (i.e., keel) and heel.[6, 7]

The stiffness properties of a prosthetic foot can have substantial effects on gait in people with LLA. For example, decreased heel stiffness (i.e., a softer heel) has been associated with prosthetic-side reduced knee flexion and reduced internal knee extensor moment during early stance,[8] increased energy return,[8–10] and faster time to foot-flat, which in turn has been associated with improved perceptions of stability.[11] Decreased prosthetic forefoot stiffness (i.e., a softer forefoot) has demonstrated a mix of effects (e.g., increased prosthetic-side energy storage and return,[8, 12, 13] increased ankle peak push-off power and work,[9, 14] an association with prosthetic side increased knee extensor and hip musculature compensation, increased intact side ankle musculature demand,[12] and increased intact limb loading[8, 10, 15]). The majority of studies that have assessed the effects of foot stiffness on gait use experimental prosthetic feet, which allow for accurate control of prosthetic foot stiffness. However, since there is limited published stiffness data across the vast majority of commercial prosthetic feet, it is challenging to translate these findings into direct clinical context.[14]

The above-mentioned effects on gait underscore the importance of matching prosthetic foot heel and forefoot stiffness properties with the abilities and goals of individual prosthetic users. Mechanical testing procedures have previously been used to quantify linear stiffness properties of prosthetic forefeet[6, 16–23] and heels,[6, 16, 17, 22, 24] but only for a limited subset of commercial feet. Further, the majority of these studies have not reported stiffness across manufacturer-defined stiffness categories within commercial foot types. To our knowledge, only two studies have included mechanical testing across consecutive stiffness categories within the same prosthetic foot model;[16, 19] one only characterized stiffness of running-specific prosthetic feet,[19] and the other only studied one foot size (i.e., 27cm).[16]

Availability of comparable forefoot and heel stiffness data would allow for a better understanding of differences not only between different commercial prosthetic feet, but also between feet of different stiffness categories and foot sizes. Therefore, the purpose of this study was to compare the linear stiffness properties of prosthetic forefeet and heels across stiffness categories and foot sizes for a range of commonly-prescribed, commercial prosthetic feet. We hypothesized that, within each foot type, calculated linear stiffness properties would increase with increasing stiffness category. Secondarily, we hypothesized that the relationship between calculated linear stiffness properties and stiffness category would be consistent across sizes of prosthetic feet.

# Materials and methods

Mechanical testing methods were used to measure the forefoot and heel linear stiffness properties of five types of commercially-available prosthetic feet, including WalkTek (Freedom Innovations; Irvine, CA), Seattle Lightfoot2 (Trulife USA; Poulsbo, WA), Vari-Flex (Össur; Reykjavik, Iceland), Rush HiPro (Proteor USA; Tempe, AZ), and AllPro 8-inch (Fillauer, Inc.; Chattanooga, TN). These commonly-prescribed prosthetic feet exhibit a range of material properties, geometries, and features, and also cover a range of user activity-levels. Sizes 27, 28, and 29 cm feet were tested for all five models, for a total of 15 groups, where each group is defined by all feet of the same type and size. Within each group (i.e., foot type and size), a

**Table 1. Manufacturer-defined stiffness categories for five types of commercial prosthetic feet.**

| Prosthetic Foot Model | Manufacturer | Sizes (cm) | Category | Medium Impact User Mean Body Weight* kg (lb.) | Maximum Allowable User Body Weight** kg (lb.) |
|---|---|---|---|---|---|
| WalkTek | Freedom Innovations; Irvine, CA | 27–29 | 1 | 52.2 (115) | 59.0 (130) |
| | | | 2 | 70.8 (156) | 81.6 (180) |
| | | | 3 | 95.7 (211) | 108.9 (240) |
| | | | 4 | 722.9 (271) | 136.1 (300) |
| Seattle Lightfoot2 | Trulife USA; Poulsbo, WA | 27–29 | 6 | 57.2 (126) | 74.8 (165) |
| | | | 7 | 79.8 (176) | 86.2 (190) |
| | | | 8 | 102.5 (226) | 115.7 (255) |
| Vari-Flex | Össur; Reykjavik, Iceland | 27–29 | 3 | 64.0 (141) | 77.1 (170) |
| | | | 4 | 73.0 (161) | 87.99 (194) |
| | | | 5 | 83.0 (183) | 100.2 (221) |
| | | | 6 | 94.3 (208) | 116.1 (256) |
| | | | 7 | 108.4 (239) | 130.2 (287) |
| Rush HiPro | Proteor USA; Tempe, AZ | 27–29 | 2 | 59.9 (132) | 66.2 (146) |
| | | | 3 | 74.4 (164) | 81.6 (180) |
| | | | 4 | 90.7 (200) | 98.9 (218) |
| | | | 5 | 108.9 (240) | 118.4 (261) |
| AllPro 8-inch | Fillauer, Inc.; Chattanooga, TN | 27–28 | C6 | 68.0 (150) | 81.2 (179) |
| | | | D7 | 91.6 (202) | 101.6 (224) |
| | | | E8 | 113.4 (250) | 124.7 (275) |
| | | 29 | D7 | 77.1 (170) | 90.3 (199) |
| | | | E8 | 102.1 (225) | 112.9 (249) |

*Averaged from the manufacturer-provided user body weight range for each stiffness category at a medium impact level.

**Maximum allowable user body weight for the category was used as the target load threshold for testing.

range of stiffness categories specified by each manufacturer for medium impact-level users between 59.0–113.4 kg (130–250 lb.) were tested. A total of 56 prosthetic feet were therefore tested (see Table 1 for details).

## Procedures

Each prosthetic foot was loaded using a six degree-of-freedom (DOF) R2000 Rotopod (Mikrolar, Inc.; Hampton, NH) robot (Fig 1). The feet were attached to the steel frame surrounding the robot (i.e., using the same double-ended female pyramid prosthetic adapter) and remained stationary throughout testing. Each foot was shod with its respective foot shell and a standardized walking shoe (MW577; New Balance; Boston, MA) to mimic clinical use. Prosthetic foot alignment was set to neutral in all planes (sagittal, coronal, and transverse). Neutral alignment was established in the transverse plane using a level positioned between two points on the shoe, while neutral in the sagittal and coronal planes was established using screws on the prosthetic foot adapter to align the shoe such that the sole of the foot was parallel to the force plate. To reduce shear forces during loading, the plantar surface of the shoe was covered in a low-friction film (i.e., ShearBan, Tamarack Habilitation Technologies, Inc.; Blaine, MN).

A 6-axis load cell (AMTI; Watertown, MA) was attached in-line with the prosthetic foot to collect force data. A motion capture system (Vicon Motion Systems Ltd.; Centennial, CO) was positioned around the sides and top of the robot's frame, with the foot at the center of the capture volume, and was used to collect displacement data. A voltage trigger was used to sync the

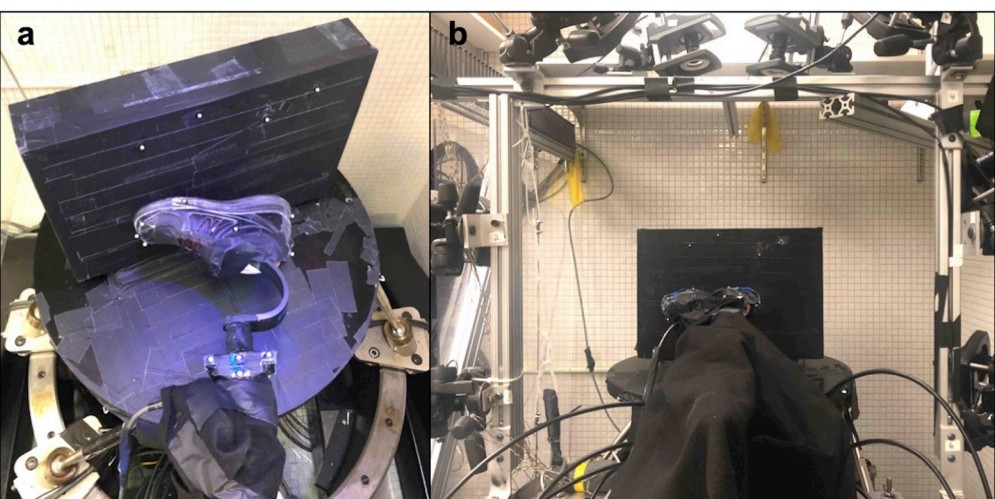

**Fig 1. Mechanical testing apparatus with example commercial prosthetic foot setup.** (a) Aerial view of prosthetic foot setup for testing with in-line load cell and reflective markers (b) view of R2000 Rotopod parallel robot with a vertically-mounted plate as a loading platform.

motion capture and load cell data. Reflective markers were placed on the loading platform, the prosthetic foot and shoe, and on the load cell, in order to define the local coordinate system.

Testing was performed at two discrete angles by orienting the R2000 loading platform relative to the foot (i.e., -15˚ to simulate independent heel loading during early stance, and +20˚ to simulate independent forefoot unloading during terminal stance) (Fig 2). Trajectory-based displacement control was used to apply quasi-static loads to the feet using custom software architecture. To initiate loading at each angle, the platform was moved along a single axis toward the mounted prosthetic foot at a fixed rate of 20 mm/s. For each trial, a minimum threshold of 50 N of normal force was maintained, per published guidelines.[17, 25, 26] The forefeet and heels were loaded to the maximum user body weight limit as determined by the manufacturer for the respective foot stiffness category (Table 1). The target load was based on overall weight limit for a category rather than weight limit for a specific activity level and therefore in some cases was higher than the user weight range indicated for a medium impact user. For each trial, the foot was loaded and unloaded for six consecutive cycles. Additional testing was completed to evaluate for any variation in stiffness between "duplicate" prosthetic feet (i.e., feet ordered from the manufacturer at different times but identical in model, size, and stiffness category) (S1 Appendix).

## Data processing

During testing, motion capture data were collected at 200 Hz and smoothed using a fourth-order Butterworth filter with a cut-off frequency of 50 Hz. Load cell data were collected at 1000 Hz and resampled at 200 Hz. Total force (N) and displacement (mm) data from the last three cycles were averaged; the first three cycles collected in each test were considered preconditioning.[17] Since a minimum force was maintained throughout testing, the data were linearly extrapolated from 50 N to 0 N using the initial ten force-displacement data points.

## Linear stiffness calculation

Calculated linear stiffness was determined using linear regression models fit to two data points (S2 Appendix) (i.e., the minimum point (50 N) and the mean target user body weight specified

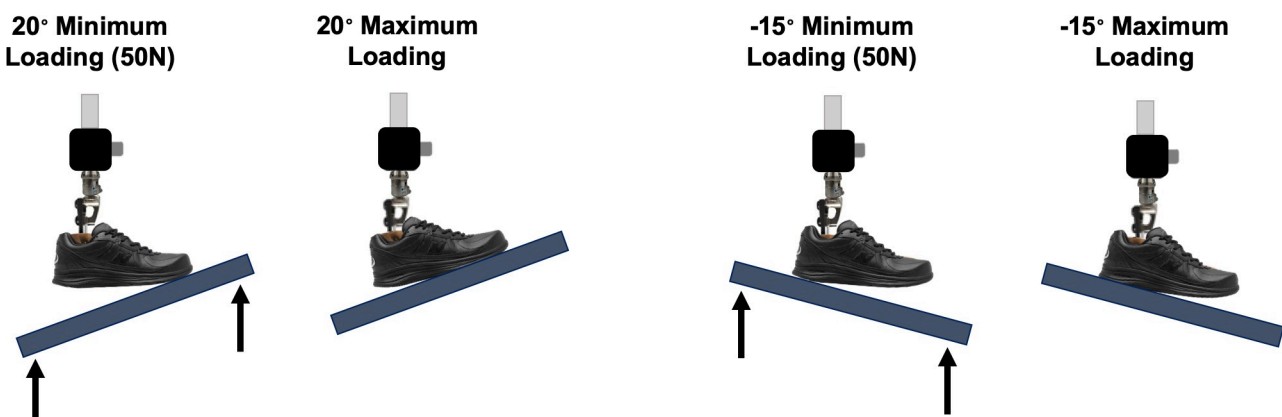

**Fig 2. Positioning of loading platform relative to prosthetic foot in mechanical testing apparatus.** Diagram of prosthetic foot mechanical testing apparatus demonstrating the positioning of the loading platform relative to the prosthetic foot for +20˚ and -15˚ testing conditions.

by each manufacturer for the respective foot stiffness category) on the force-displacement curve (Table 1). Stiffness was calculated using data from the unloading portions of the forefoot force-displacement curves and from the loading portions of the heel curves. The selected portions of the curves were chosen to reflect stiffness behavior representative of foot loading during the respective phase of gait (i.e., loading of the heel during early stance and unloading of the forefoot during late stance).

## Relative difference between low-load stiffness to high-load stiffness

While a single value for linear stiffness is an indicator of overall stiffness, there are different degrees of nonlinearity across prosthetic feet (e.g., different stiffnesses at lower and higher loads). Therefore, we created an additional method to quantify the relative difference between low-load stiffness compared to high-load stiffness. Two measurements were used to establish a percentage of difference in stiffness behavior during low loads compared to during high loads: high-load stiffness and low-load stiffness (S2 Appendix). A greater percentage difference indicates a larger magnitude of change between the stiffness experienced at low loads compared to that at high loads (i.e., greater nonlinearity).

## Statistical methods

Linear mixed effects regression was used to assess associations between calculated forefoot or heel stiffness (dependent variable) and stiffness category (independent fixed effect) with foot size as a fixed effect covariate, modeled as categorical (S3 Appendix). Foot type and foot type by stiffness category interaction were random effects. Stiffness category across feet was scaled so that a value of 1 represented minimum stiffness category (i.e., rated for 59.0 kg (130 lb.)) and 5 represented maximum stiffness category (i.e., rated for 113.4 kg (250 lb.)). Scaling was distributed equally within this range depending on the number of manufacturer stiffness categories for each foot type and size. Likelihood ratio tests were carried out to test the hypothesis of no association between calculated stiffness and stiffness category (i.e., the stiffness category coefficient is equal to zero) to test for variability in this association across foot types (by testing the significance of the foot type by stiffness category interaction term). To address our secondary hypothesis, the influence of foot size on the association between calculated stiffness and stiffness category was examined by testing the significance of stiffness category by size interaction terms. Results are summarized with the stiffness category model coefficient (representing

**Table 2. Calculated linear stiffness properties (N/mm) by manufacturer-defined stiffness categories.**

| Commercial Foot Type | Stiffness Category | Forefoot Stiffness (N/mm) | | | Heel Stiffness (N/mm) | | |
|---|---|---|---|---|---|---|---|
| | | 27cm | 28cm | 29cm | 27cm | 28cm | 29cm |
| WalkTek | 1 | 21.5 | 21.1 | 20.6 | 40.7 | 39.4 | 39.4 |
| | 2 | 26.4 | 25.6 | 25.6 | 44.1 | 42.6 | 46.3 |
| | 3 | 38.4 | 35.7 | 36.7 | 60.4 | 58.3 | 65.4 |
| | 4 | 46.1 | 42.4 | 44.6 | 63.6 | 70.6 | 69.3 |
| Seattle Lightfoot2 | 6 | 26.4 | 26.3 | 20.7 | 55.0 | 48.5 | 51.8 |
| | 7 | 28.4 | 32.3 | 28.3 | 50.9 | 58.9 | 64.9 |
| | 8 | 37.4 | 37.8 | 34.4 | 57.8 | 64.5 | 65.7 |
| Vari-Flex | 3 | 23.1 | 27.3 | 24.0 | 32.4 | 25.5 | 28.3 |
| | 4 | 26.7 | 31.9 | 25.8 | 35.6 | 34.7 | 34.7 |
| | 5 | 29.1 | 36.4 | 31.8 | 37.5 | 36.5 | 36.2 |
| | 6 | 33.6 | 39.7 | 35.4 | 47.4 | 42.4 | 43.6 |
| | 7 | 38.5 | 46.5 | 40.6 | 45.4 | 48.7 | 50.0 |
| Rush HiPro | 2 | 19.8 | 26.5 | 23.5 | 40.8 | 28.4 | 29.8 |
| | 3 | 21.4 | 28.3 | 24.8 | 47.7 | 30.3 | 35.1 |
| | 4 | 27.9 | 43.5 | 26.2 | 48.9 | 40.0 | 43.8 |
| | 5 | 34.8 | 33.6 | 30.9 | 53.4 | 47.7 | 56.1 |
| AllPro, 8in | C6 | 18.1 | 16.4 | - | 41.1 | 44.5 | - |
| | D7 | 22.2 | 22.7 | 18.7 | 46.7 | 44.2 | 45.1 |
| | E8 | 27.4 | 28.1 | 24.8 | 55.7 | 57.1 | 53.1 |

the change in calculated stiffness per 1 unit increase in stiffness category) ± standard error (SE), 95% confidence intervals (CI) and marginal $R$-squares ($R^2$). Analyses were carried out using R 3.6.2, and packages tidyverse, lme4, emmeans and MuMIn.[27]

## Results

### Linear stiffness

Calculated linear stiffness across all foot types, sizes, and stiffness categories ranged from 17.3–44.4 N/mm for forefeet and from 28.7–67.8 N/mm for heels (Table 2). Overall, forefoot and heel stiffnesses were both significantly associated with stiffness category ($R^2$ = 0.54 and 0.38, respectively, $p$ = .001) (Table 3, Figs 3 and 4). Slopes for forefoot and heel stiffness varied significantly by commercial foot type (foot type by stiffness category interaction $p < .01$) (Table 3, Figs 3 and 4). Overall, slopes did not differ by foot size (stiffness category by size interaction $p$ = .80).

**Table 3. Summary statistics from linear mixed effects regression models of calculated stiffness on stiffness categories and foot sizes.**

| | Mean change in calculated stiffness between categories | | | Variability by foot type | | |
|---|---|---|---|---|---|---|
| | Slope ± SE | (95% CI) | $p^*$ | Slope Range | SD (95%CI) | $p$† |
| Forefeet | 3.7 ± 0.7 | (2.2, 5.1) | .001 | 2.1–5.6 | 1.4 (0.6, 2.9) | < .001 |
| Heels | 4.6 ± 0.9 | (2.7, 6.5) | .001 | 3.0–7.1 | 1.8 (0.7, 3.7) | < .01 |

*significance of stiffness category coefficient

†significance of stiffness category by foot type interaction

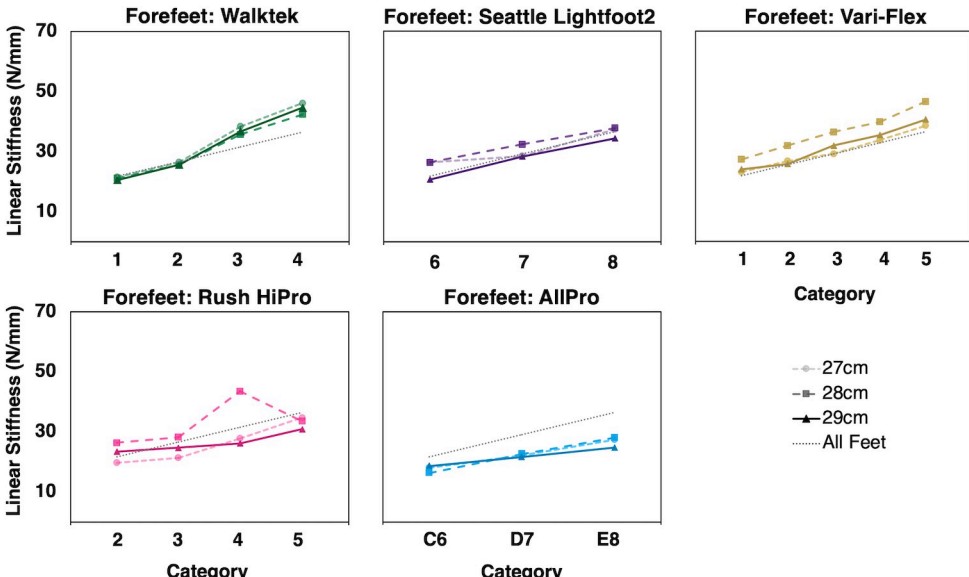

**Fig 3. Calculated linear stiffness of prosthetic forefeet by scaled stiffness category (min 1, max 5).** Five types of commercial prosthetic feet with three different foot sizes (i.e., 27-29cm) tested for each stiffness category (for a total of 15 groups, where group is defined by all feet of the same type and size). Dashed lines indicate the regression slope averaged across all foot types and sizes from linear mixed effects regression of calculated stiffness on stiffness category and size, with random effects for foot type and foot type by stiffness category interaction. Calculated linear stiffness was significantly associated with scaled stiffness category ($p = 0.001$), although depending on foot type, variability in stiffness across foot sizes and across consecutive stiffness categories is observed ($p<0.01$).

On average across foot sizes and all foot types, for a one unit increase in scaled foot stiffness category (i.e., 1–5 from minimum category to maximum category), there was a 3.7 ± 0.7 N/mm (CI: 2.2, 5.1) estimated increase in calculated forefoot stiffness. Similarly, in the heel data, for a one unit increase in scaled foot stiffness category, there was a 4.6 ± 0.9 N/mm (CI: 2.7, 6.5) estimated increase in calculated stiffness (Table 3).

While on average, stiffnesses increased with stiffness category, there were inconsistencies between the expected and measured incremental stiffness changes (i.e., relative incremental change in stiffness across foot stiffness categories within each foot type). Inconsistencies between expected and measured stiffness changes, either between categories or between sizes, were found for the vast majority of groups of tested forefeet and heels (Table 2, Figs 3 and 4); there were numerous instances where the magnitude of stiffness change across foot stiffness categories within one foot type varied considerably (e.g., one foot type demonstrated an average forefoot stiffness increase of 4.8 ± 0.3 N/mm from category 1 to 2 across foot sizes, but demonstrated twice the average increase between categories 2 and 3 (i.e., 11.1 ± 0.9 N/mm) in all foot sizes), and there were even instances where a decrease in calculated stiffness was found as foot stiffness category increased (Table 2).

## Relative difference between low-load stiffness and high-load stiffness

The degree of nonlinearity (difference between low-load stiffness and high-load stiffness) varied substantially across feet tested; relative change in stiffness ranged from 28–86% for the forefeet and from 52–94% for the heels, across all feet (Table 4). It was also noted that some types of feet had substantially different percentage differences across sizes, while other types of feet had very consistent percentage differences across foot sizes and categories (i.e., consistent nonlinearity observed in force-displacement data).

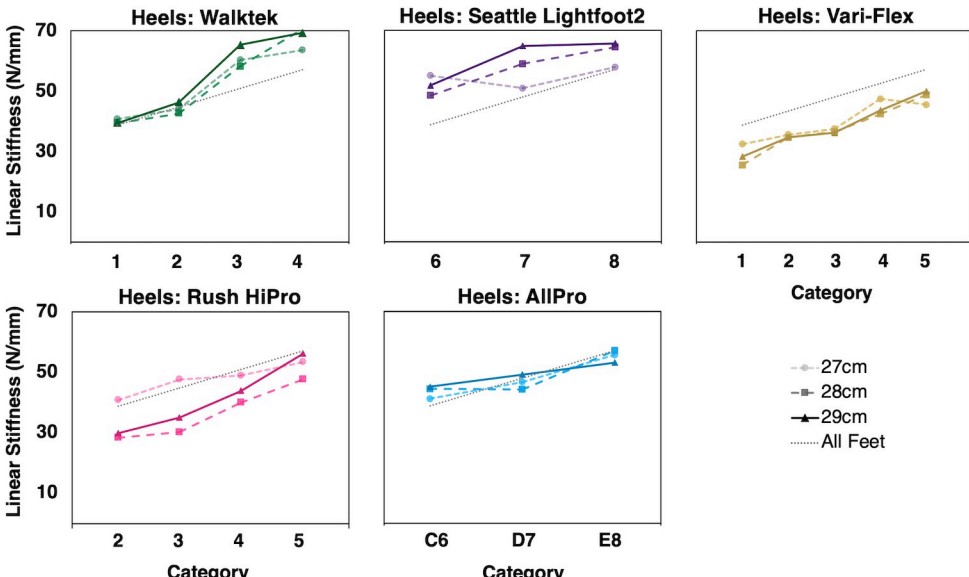

**Fig 4. Calculated linear stiffness of prosthetic heels by scaled stiffness category (min 1, max 5).** Five types of commercial prosthetic feet with three different foot sizes (i.e., 27-29cm) tested for each stiffness category (for a total of 15 groups, where group is defined by all feet of the same type and size). Dashed lines indicate the regression slope averaged across all foot types and sizes from linear mixed effects regression of calculated stiffness on stiffness category and size, with random effects for foot type and foot type by stiffness category interaction. Calculated linear stiffness was significantly associated with scaled stiffness category ($p = 0.001$), although depending on the type of foot, variability in stiffness across foot sizes and across consecutive stiffness categories is observed ($p<0.01$). On average, heel stiffness values were greater than forefoot stiffness values, across all foot types, which may be attributed to the shorter lever arm of the heel compared to that of the forefoot.

## Force-displacement curves

Forefoot (S4 Appendix) and heel (S5 Appendix) force-displacement curves demonstrated non-linear force-displacement behaviors under load. Despite the overall correlation between calculated stiffness and stiffness category, there were instances in which the displacement experienced throughout the loading cycle by consecutive categories did not match the expected pattern (e.g., higher categories did not always experience less displacement as would be expected with a stiffer foot). These variations in displacement that occurred with loading or unloading were identified by the order of force-displacement curves for all stiffness categories within each size and foot type combination. The relative position of force-displacement curves was inconsistent with the expected order of foot stiffness categories in the majority of groups of forefeet and heels (i.e., consecutive categories were not ordered as expected) (Fig 5B).

## Discussion

This study compared stiffness properties of a variety of commercial prosthetic feet (i.e., forefoot and heel regions) across manufacturer-defined foot stiffness categories and foot sizes. Across all types of prosthetic feet, increased calculated forefoot and heel stiffness was, on average, significantly correlated with increased stiffness category, and the relationship between calculated stiffness properties and stiffness category did not vary by foot size. However, inconsistencies existed between the expected and calculated stiffness changes, both across categories and between sizes. Examples of these inconsistencies occurred in the vast majority of forefoot and heel groups tested. More than half of the forefeet and over 90% of the heels studied exhibited inconsistencies across consecutive stiffness categories in the displacement

**Table 4. Relative percentage differences of low-load compared to high-load stiffness behavior.**

| Commercial Foot Type | Stiffness Category | Forefoot Relative Difference in Low-Load to High-Load Stiffness | | | Heel Relative Difference in Low-Load to High-Load Stiffness | | |
|---|---|---|---|---|---|---|---|
| | | 27cm | 28cm | 29cm | 27cm | 28cm | 29cm |
| WalkTek | 1 | 83% | 85% | 86% | 54% | 52% | 52% |
| | 2 | 84% | 85% | 88% | 73% | 64% | 67% |
| | 3 | 78% | 83% | 82% | 71% | 67% | 61% |
| | 4 | 83% | 83% | 85% | 72% | 78% | 72% |
| Seattle Lightfoot2 | 6 | 43% | 70% | 76% | 73% | 77% | 66% |
| | 7 | 78% | 79% | 82% | 75% | 81% | 74% |
| | 8 | 79% | 81% | 83% | 70% | 82% | 76% |
| Vari-Flex | 3 | 62% | 55% | 78% | 68% | 75% | 76% |
| | 4 | 65% | 61% | 64% | 74% | 70% | 73% |
| | 5 | 66% | 48% | 64% | 80% | 77% | 73% |
| | 6 | 67% | 56% | 61% | 72% | 74% | 71% |
| | 7 | 75% | 61% | 69% | 81% | 74% | 73% |
| Rush HiPro | 2 | 60% | 61% | 53% | 91% | 92% | 94% |
| | 3 | 28% | 66% | 60% | 93% | 92% | 93% |
| | 4 | 67% | 70% | 67% | 93% | 92% | 95% |
| | 5 | 65% | 70% | 70% | 93% | 89% | 93% |
| AllPro, 8in | C6 | 71% | 42% | - | 65% | 63% | - |
| | D7 | 68% | 36% | 38% | 66% | 79% | 67% |
| | E8 | 69% | 40% | 38% | 69% | 83% | 70% |

observed throughout the loading cycle. One potential explanation for this observation is that manufacturer-defined user body weight ranges varied somewhat across stiffness categories within types of feet (e.g., some stiffness categories were intended for a broader range of user

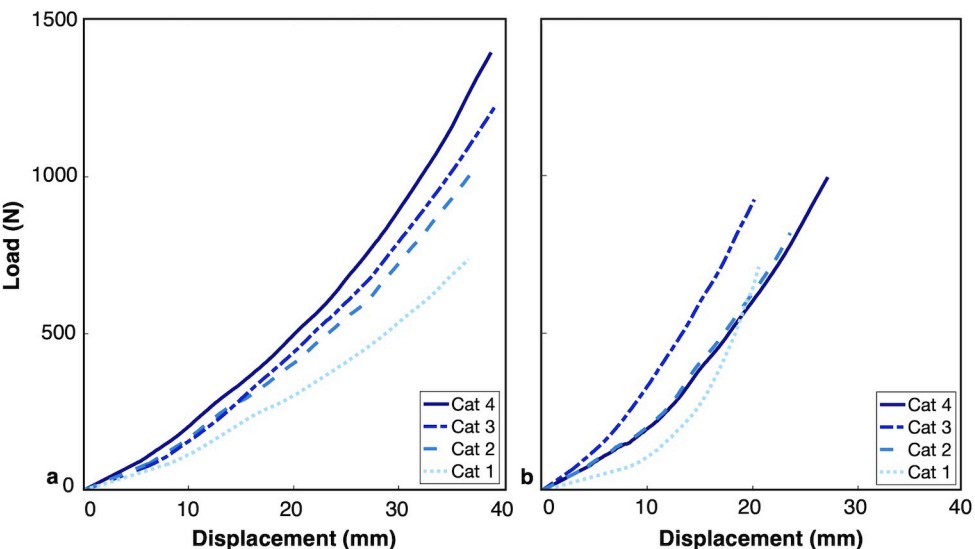

**Fig 5. Illustrations of example force-displacement curves generated from mechanical testing of prosthetic foot forefeet or heels.** Force-displacement plots demonstrate examples of (a) no inconsistencies observed in the order of consecutive category force-displacement curves, and (b) several inconsistencies observed between consecutive categories (e.g., cat 3 is stiffer than cat 4), multiple categories have similar data (i.e., cat 2 and cat 4), and there are differences in relative low-load to high-load stiffness behavior across categories (e.g., cat 1 compared to cat 2).

body weights than others). While differing incremental increases in user body weight between consecutive categories could result in inconsistent stiffness increases (e.g., larger changes in stiffness may be expected between categories with broader weight ranges), this would not explain the observed instances of force displacement curve overlapping across consecutive stiffness categories, nor would it explain the instances of apparent decrease in stiffness with an increase in stiffness category. Overlapping of consecutive categories could be explained if certain heel or forefoot components are used across multiple stiffness categories. However, there are no data published by foot manufacturers to contextualize these results. Furthermore, these findings of inconsistencies in calculated stiffness across stiffness categories suggest that manufacturer-reported stiffness categories alone may not provide sufficient information for clinicians to choose an optimal prosthetic foot for a given individual prosthetic user.

The few previous studies that included mechanical testing of commercial prosthetic feet of varying stiffness categories reported results congruent with inconsistencies found in the current study. In one study evaluating the stiffness properties of four different types of running-specific commercial prosthetic feet, the calculated stiffness values were found to increase with increasing stiffness category for all models of feet, but incremental changes in calculated stiffness values were not equal across categories.[19] Another recent study of a variety of size 27cm prosthetic feet reported instances in which higher-category commercial prosthetic feet had lower stiffnesses than lower-category feet of the same type.[16]

While the ISO standards (e.g., 22675) specify mechanical testing parameters for safety testing, they do not establish criteria or recommendations for determining prosthetic forefoot and heel stiffnesses.[26] Therefore, a range of approaches have been used to quantify stiffness properties in previous studies of prosthetic feet using mechanical testing.[6, 16–19, 21, 22, 24, 26, 28] A common method is using a linear regression across the full load range (i.e., data from the entire force-displacement curve) to estimate the slope.[17, 19, 21, 22] Calculated linear stiffness in this manner is appealing in that it is an easily understood numerical value that would be easy for clinicians to use when comparing prosthetic feet. Several studies also have used a "functional" or instantaneous stiffness estimate, in which linear regression was fit to a smaller region of the force-displacement curve, at a particular load level (e.g., approximating near body weight).[6, 16, 18] However, distilling the information from each force-displacement curve to a single value of stiffness may be better suited for a foot with linear behavior, and is likely less representative of many contemporary prosthetic feet. Since many prosthetic feet have curvilinear behavior, one study described using a 2nd order polynomial, in addition to calculating the linear stiffness of feet across the full curve, as it provided a better representation of the nonlinear force-displacement curve.[19] However, equations like 2nd order polynomials are more difficult to interpret and thus may have less intuitive meaning to describe stiffness behavior than linear slopes.

Calculated stiffness only sufficiently captures linear behavior, and therefore does not provide a detailed picture of foot behavior across categories and sizes. All forefeet and heels in the present study exhibited nonlinear mechanical behavior during loading similar to the findings from previous studies.[6, 16–19, 22, 24] While the numerical linear stiffness values are important for quantifying stiffness behavior en masse, more detailed observations can be gleaned from the force-displacement curves. Given that there are different degrees of nonlinearity across feet, we calculated both the linear stiffness as well as the relative difference between low load stiffness and high load stiffness. There were cases in which stiffness behavior changed substantially during loading (i.e., curves showed low stiffness behavior at low loads, followed by a sharp increase in stiffness at an inflection point, and higher stiffness behavior for the remainder of loading). In some instances, the change in stiffness behavior was inconsistent across categories or foot sizes. However, in other cases the large relative percentage change in stiffness

was observed across all foot sizes and is likely due to the presence of an elastomeric bumper as part of the forefoot or heel components. The nonlinearity observed on the force-displacement curves is consistent with observations for similar designs of commercial feet tested in previous studies,[16] and is likely due to the elastic properties of materials (e.g., composites and elastomers), geometries of prosthetic feet, and shifting center of pressure during loading. However, consistency in nonlinearity of force-displacement curves across size and category was expected within a given foot type.

In addition to stiffness quantification, methods of mechanically testing prosthetic feet vary across studies. Although there have been numerous previous studies that have mechanically tested prosthetic feet, we identified three such studies that were similar to the current study in the types of prosthetic feet (size, stiffness category, and model (i.e., Rush HiPro,[18] AllPro, [18] Vari-Flex,[16] and Seattle Lightfoot2[16, 17])) and pylon progression angles tested. In the current study we used an R2000 Rotopod for mechanical testing. Womac et al. used the same mechanical testing equipment,[16] whereas Major et al. used a materials testing machine and sine plate (as described in ISO standard 10328),[17] and Koehler et al. used a materials testing machine with the foot fastened to an aluminum bar.[18] While in the current study we tested feet at -15˚ and +20˚ sagittal pylon progression angles, Womac et al. tested at 15 different sagittal pylon progression angles ranging from -15˚ to +30˚ (including the two pylon progression angles used in the current study),[16] Major et al. tested at -15˚ and +20˚ (the two angles tested in the current study) as well as a level loading surface,[17] and Koehler et al. tested only at a +20˚ pylon progression angle.[18] Target loading thresholds and the number of loading cycles also varied by study. The current study used the manufacturer-specified maximum allowable user body weight for each foot stiffness category when determining loading thresholds. Six loading cycles were performed per foot-pylon-progression-angle combination, with a 50 N minimum load. Womac et al. used the vertical ground reaction force from previously tested human subjects (people with transtibial amputation) walking in a gait laboratory as pylon-progression-angle-specific target loads and one cycle per foot-pylon-progression-angle combination.[16] Major et al. used a target load of 1230 N with 50 N of preloading and two cycles per foot-pylon-progression-angle combination.[17] Koehler et al. individualized target loads by using the body weight of three individuals with transtibial amputation walking with weighted vests.[18] The differences in corresponding calculated linear stiffness values between the current study and these previous studies were generally small (see S6 Appendix which details differences in forefoot and heel calculated stiffness across studies for comparable prosthetic feet and pylon progression angles), suggesting that the current testing methods are comparable with previous methods.

There are several limitations to this study. The prosthetic feet studied are only a subset of the hundreds of commercially-available foot models, and therefore cannot represent all other feet. However, the prosthetic foot models were chosen to provide a variety of foot designs that would be commonly-prescribed for users from a range of activity levels. Another potential limitation is that the loading rate used (20 mm/s) is likely slower than physiological loading rates during walking. This rate was used because it was the fastest possible with the testing equipment and resulted in a faster loading rate than previous mechanical testing studies, in terms of force (N/s).[6, 16–19] Additionally, quasi-static loading performed at two, sagittal-plane pylon progression angles is not representative of loading throughout the full gait cycle, although it is consistent with previous studies.[6, 17, 18, 22, 28] Despite these limitations, mechanical testing allows for the collection of user-independent data and avoids confounding variables due to variation across individual's gait. Finally, linear stiffness is not the only property of prosthetic feet that may affect gait. While linear stiffness is an important feature, especially when comparing across stiffness categories of prosthetic feet, future studies should develop methods of

quantifying other prosthetic foot mechanical properties that take into account nonlinear behaviors.

## Conclusions

The numerous inconsistencies found in calculated stiffnesses across prosthetic foot stiffness categories suggest the importance of more standardized procedures for testing, analyzing, and reporting prosthetic foot mechanical testing data. Since manufacturers do not typically report prosthetic forefoot and heel stiffness values, the findings from this study can help inform prosthetic foot prescription by allowing clinicians to better match prosthetic users' individual abilities and mobility goals with prosthetic foot stiffness data (at least for the five types of feet assessed in the current study). Future work is warranted to measure stiffnesses (and other mechanical properties) of a larger variety of commercial prosthetic feet.

## Supporting information

**S1 Appendix. Testing of duplicate foot models.** Testing was completed to evaluate variation in stiffness between "duplicate" prosthetic feet (i.e., feet ordered from the manufacturer at different times but identical in model, size, and stiffness category). Pairs of duplicate size 27 cm feet at two stiffness categories (i.e., corresponding to user weights of 68.0 and 90.7 kg (150 and 200 lb.)) of each foot type were tested. A total of 10 sets of duplicate feet were therefore compared. Testing was performed at pylon progression angles of -10° and +20° to simulate heel and forefoot loading, respectively. The experimental setup and all other procedures otherwise matched those described in the manuscript. Calculated linear stiffness was very similar (mean difference of 2.4 ± 1.3%) between all pairs of duplicate feet, for all foot types and stiffness categories (Fig A). The data between duplicate feet were similar both in calculated linear stiffness values and in the features observed in the force-displacement curves, suggesting repeatability in manufacturing across multiple samples of the same foot model. Similar results were observed in a previous study which tested multiple samples of a commercial foot [21].
(DOCX)

**S2 Appendix. Calculation of linear stiffness and determination of relative difference between low-load stiffness and high-load stiffness.** Example force-displacement data including highlighted areas representing the low and high-load areas of the curve used to determine relative difference in stiffness. Force-displacement curves also demonstrate method of linear stiffness calculation (i.e., linear stiffness used to quantify stiffness for each foot).
(DOCX)

**S3 Appendix. Statistical models used in linear mixed effects analyses.**
(DOCX)

**S4 Appendix. Force-displacement curves for forefeet, arranged by prosthetic foot type and size.** All tested stiffness categories within each prosthetic foot type and size are shown. Force and displacement data from the last three loading cycles were averaged (i.e., mean line shown for each foot) and standard deviation across cycles is demonstrated by shaded regions around each force-displacement curve. Unloading portions of the forefoot curves are shown. Since a minimum force was maintained throughout testing, the data were linearly extrapolated to 0 N to provide an estimate for displacement at low loads. This was completed using a linear model fit to the initial ten force-displacement data points to estimate the earliest linear stiffness behavior, thus data shown below 50 N of load has been extrapolated.
(DOCX)

**S5 Appendix. Force-displacement curves for heels, arranged by prosthetic foot type and size.** All tested stiffness categories within each prosthetic foot type and size are shown. Force and displacement data from the last three loading cycles were averaged (i.e., mean line shown for each foot) and standard deviation across cycles is demonstrated by shaded regions around each force-displacement curve. Loading portions of the heel curves are shown. Since a minimum force was maintained throughout testing, the data were linearly extrapolated to 0 N to provide an estimate for displacement at low loads. This was completed using a linear model fit to the initial ten force-displacement data points to estimate the earliest linear stiffness behavior, thus data shown below 50 N of load has been extrapolated.
(DOCX)

**S6 Appendix. Forefoot and heel calculated stiffness comparison across studies for comparable prosthetic feet and pylon progression angles.**
(DOCX)

**S7 Appendix. Force-displacement data from prosthetic foot mechanical testing.**
(XLSX)

## Acknowledgments

We would like to thank Jane Shofer, M.S. for performing the statistical data analysis.

## Author Contributions

**Conceptualization:** Elizabeth G. Halsne, Andrew H. Hansen, Brian J. Hafner, David C. Morgenroth.

**Data curation:** Elizabeth G. Halsne.

**Formal analysis:** Elizabeth G. Halsne.

**Funding acquisition:** David C. Morgenroth.

**Investigation:** Anne T. Turner, Elizabeth G. Halsne.

**Methodology:** Anne T. Turner, Elizabeth G. Halsne, Joshua M. Caputo, Carl S. Curran.

**Project administration:** David C. Morgenroth.

**Software:** Anne T. Turner, Elizabeth G. Halsne.

**Supervision:** David C. Morgenroth.

**Visualization:** Anne T. Turner, Elizabeth G. Halsne.

**Writing – original draft:** Anne T. Turner, Elizabeth G. Halsne, David C. Morgenroth.

**Writing – review & editing:** Anne T. Turner, Elizabeth G. Halsne, Joshua M. Caputo, Carl S. Curran, Andrew H. Hansen, Brian J. Hafner, David C. Morgenroth.

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
