## [Decision Letter · Decision Letter 0]

27 Aug 2021

PONE-D-21-22884

Prosthetic forefoot and heel stiffness across consecutive foot stiffness categories and sizes

PLOS ONE

Dear Dr. Morgenroth,

Thank you for submitting your manuscript to PLOS ONE. After careful consideration, we feel that it has merit but does not fully meet PLOS ONE’s publication criteria as it currently stands. Therefore, we invite you to submit a revised version of the manuscript that addresses the points raised during the review process.

We look forward to receiving your revised manuscript.

Kind regards,

Arezoo Eshraghi, Ph.D.

Academic Editor

PLOS ONE

Journal Requirements:

[This research is a project of the Seattle Institute for Biomedical and Clinical Research (https://www.sibcr.org/) supported by the Office of the Assistant Secretary of Defense for Health Affairs, through the Orthotics and Prosthetics Outcomes Research Program (https://cdmrp.army.mil/oporp/default) under Award No. W81XWH-16-1-0569 (PI: D.C.M.). Opinions, interpretations, conclusions and recommendations are those of the author and are not necessarily endorsed by the Department of Defense. The U.S. Army Medical Research Acquisition Activity, 820 Chandler Street, Fort Detrick MD 21702-5014 is the awarding and administering acquisition office. This material is the result of work supported with resources and the use of facilities at the VA Puget Sound Health Care System, Seattle, Washington (https://www.pugetsound.va.gov/). The contents do not represent the views of the United States Department of Veterans Affairs or the United States Government.

This study was not funded by commercial companies and no authors received funding from commercial companies (except that co-authors J.M.C. and C.S.C. are employed by Human Motion Technologies LLC d/b/a Humotech). Humotech was not a funder for this study, and the funder (Department of Defense) had no role in study design, data collection and analysis, decision to publish, or preparation of the manuscript.]    

We note that one or more of the authors are employed by a commercial company: Human Motion Technologies LLC d/b/a Humotech). 

Reviewers' comments:

Reviewer's Responses to Questions

**Comments to the Author**

1. Is the manuscript technically sound, and do the data support the conclusions?

Reviewer #1: Yes

Reviewer #2: Yes

Reviewer #3: Yes

2. Has the statistical analysis been performed appropriately and rigorously? 

Reviewer #1: Yes

Reviewer #2: Yes

Reviewer #3: I Don't Know

3. Have the authors made all data underlying the findings in their manuscript fully available?

Reviewer #1: Yes

Reviewer #2: Yes

Reviewer #3: Yes

4. Is the manuscript presented in an intelligible fashion and written in standard English?

Reviewer #1: Yes

Reviewer #2: Yes

Reviewer #3: Yes

5. Review Comments to the Author

Reviewer #1: - Is the manuscript technically sound, and do the data support the conclusions?

Yes, the research methods seem sound and in line with ISO testing standards. The only question I have about the methods is the addition of a standard shoe for testing, as this just adds a layer of unnecessary material between the device and the surface. From a clinical standpoint it may make sense, but because this is really a characterization of the devices, I feel that the standard shoe is unnecessary but also isn't a problem.

- Has the statistical analysis been performed appropriately and rigorously?

yes, the statistical analysis was clearly described and looks appropriate.

- Have the authors made all data underlying the findings in their manuscript fully available?

yes, the supplemental files contain graphs of all of the force/displacement curves from all of the trials of both the forefoot unloading and heel loading. Data is only presented in a graph, which is sufficient, but possibly the raw data could be added in a file for others to use, particularly for doing further analysis of non-linear behaviors.

-Is the manuscript presented in an intelligible fashion and written in standard English?

yes, the English is fluid and highly readable. The phrasing is clear and descriptions understandable. Remarkably I don't have any typos or grammatical changes.

- *5. Review Comments to the Author

This was exactly the information I was recently looking for and found quite lacking in the literature so I am really happy to see someone working on this. The authors have selected a wide range of ESR ankles to test in a wide range of sizes and weight classes to get a general idea of the change in stiffness as these variables changed. This is a really good data set to have available for everyone interested in current ankle design stiffness and the relative differences between sizes, weight and manufacturer. Overall I think it is a nice study that certainly should be published.

Overall the method is clear and the results are understandable, but there are some outliers in the data that I would hope are not due to the setup. It seems that each device was run once, so variation due to the setup can't really be determined.

These are the type of data I wish the manufacturers would publish, and I don't know if the authors have reached out to any of them to see if they would provide numbers before or during the study. Another comment is that spring constants for mass produced steel springs generally don't have very accurate spring constants. An average compression die spring usually has a +/- 20% tolerance, while a 'high precision die spring' tends to have something like +/-10%. Although some carbon springs can be held to 2%. I don't think the manufacturers ever state this kind of value or what the reject values would be for their production, but it is possible that there is large variation even within the same device size and category which this study wouldn't be able to capture.

One comment about these studies of linear stiffness estimation of the prosthesis is that it is a bit hard to compare to joint torque data that is generally produced from biomechanical studies. I don't know if there is a way to easily convert from the forces to a joint angle approximation, as it would require a bit more information about the way the prosthesis was deforming and the geometry of the device. Defining exactly where on each of the devices contact was made with respect to the geometry of the device is maybe a bit too much to report here, but suggestions would be welcome.

-I wish there was a better figure showing the direction of the forces applied to the prosthesis including the positions the displacement is measured from. This would make it clear that it is -15 and +20 degrees from the vertical. Similar to diagrams shown in ISO 10328:2016.

-Please convert and report all values in SI, even though I know the references you are using are in pounds and pound force.

Reviewer #2: This paper presents the mechanical characterization of a number of prosthetic feet, which I believe will be valuable for researchers, and potentially clinicians as well.

Major:

The statistics are not described clearly, or with adequate precision. Rather than describing something like “the significance of the slope of change of delta y over delta x1,” just describe “the significance of the x1 coefficient,” or “significance of the x1-x2 interaction term”

While obtaining so many feet to test is admirable, I worry that the total content is relatively low for a “full report.” I do not know if PLOS has a short communication format, but I would recommend this more as a short communication.

I strongly recommend reworking each of figures 2 and 3 into single-plot figures, using color to distinguish between foot type. As the effect of foot length is marginal, I suggest averaging stiffness across foot-length, instead of showing individual lines for each foot length. Thus each figure consists of one plot, which consists of five lines, which span stiffness categories. However, the scaling of the categories also presents problems. The numbers are already rather meaningless by themselves, and the mapping to [1,5] is arbitrary enough that I suggest removing it. For this to be simple for clinicians to use as a reference, it makes sense to avoid any additional mapping (or reading, really) that they need to do to understand which foot is which. Perhaps you could change the x-axis into recommended weight (or “medium impact user mean body weight”), and add small numbers next to the points on the line denoting the manufacturer-specific “category”. This would cleanly describe all the most relevant data: difference in keel/heel stiffness between models and between categories (and actual category numbers). In the supplementary information, you could include data as it’s currently shown in Figs 2 and 3 (with foot length included), but perhaps with non-scaled category on the x-axis.

It’s unclear why loading of the heel and unloading of the keel are used in the analysis. Why is heel loading more important than heel unloading, and vice versa for the keel? It seems preferable to measure both the same way. It may also be beneficial to calculate hysteresis, though at such slow loading and unloading rates that may not be realistic.

Minor:

Figure 1 should contain new images, shot in improved lighting, with improved resolution, and oriented in intuitive ways

-Citations 4, 5, 7 don’t seem to fit the sentence they are supporting particularly well

-“…and internal knee flexion moment during early stance” –do you mean increased internal knee flexion?

-“Decreased prosthetic forefoot stiffness 31 (i.e., a softer forefoot) has demonstrated a mix of potentially beneficial effects (e.g., increased prosthetic-side energy storage and return as well as increased ankle peak push-off power and work” –This is somewhat controversial, as many people don’t believe that increasing prosthesis energy storage and return is always beneficial (specifically, energy storage and return continues to increase as prosthesis stiffness is decreased to unreasonably low values). I suggest removing this argument.

-“Further, the majority of these studies have not reported linear stiffness across manufacturer-defined stiffness categories within commercial foot types” --it’s not clear what is important about “linear stiffness” in this context… perhaps remove “linear” here?

How was “neutral” alignment found/defined?

“a first-order polynomial fit” –maybe just linear fit?

“(i.e., the stiffness experienced by an average user during heel strike or forefoot push-off).” –At heel strike, I would actually define the load as being exactly 0, but increasing (the moment the heel begins to strike is different than the peak load experienced during the heel strike phase). Please clarify.

Why is foot size a categorical variable? It is a measurable quantity that is definable across models, and you will lose power by treating it as a categorical variable

The statistical test could be described more precisely. For example, “the mean slope of change (i.e., the relationship between amount of stiffness change per stiffness category)” –could this be simplified to: “the coefficient relating category to stiffness”

“Effect modification” –what does this mean?

“slope of change” is not a term I have come across, and seems like an odd mix of phrases.

I recommend writing out the equation of the linear mixed effects model used, and referencing the parameters by their symbols.

“although variability in stiffness across foot sizes and across consecutive stiffness categories is observed, depending of the type of foot (p<0.01).” –This is a bit unclear to me, consider rewording.

It’s important to discuss that the moment arm of the heel spring is much shorter, so that even though it is stiffer linearly, the angular stiffness about an approximate ankle joint, or about an instantaneous center of rotation, will not be the same.

Please discuss why a nonlinearity exists in the force-displacement curves. Is the center of pressure moving backwards? Is it material nonlinearity? Other geometric nonlinearity?

Figure 4 doesn’t illustrate much new. It would actually be helpful to show a similar set of plots earlier, for readers to better understand the low and high stiffness portions.

For figures that have curves indicating some sort of gradation of a continuous variable (for example in the supplementary figures), I recommend replacing the random colors with varying shades or darkness of a single color, so that readers don’t have to consistently reference the legend to see if the trend is as expected.

I recommend referencing the supplementary material more explicitly.

I recommend mentioning manufacturing tolerances as a source of this error, and noting that the testing of individual feet is subject to this variability.

Reviewer #3: In this manuscript a number of prosthetic feet from five manufacturers in different sizes and stiffness categories has been studied by the authors to compare the linear stiffness properties of prosthetic forefeet and heels. The study has been conducted scientifically and quite all the questions which come in mind during reviewing this manuscript, has been answered in the context. Although, since all the experiments on a prosthetic foot has been repeated on the same prosthetic foot (loading and unloading for six consecutive cycles on the same foot) and the experiments has not been repeated on another prosthetic foot with the same characteristic (manufacturer, size, stiffness category), some unexpected results, like a decrease in measured stiffness as foot stiffness category increased, could happen because of the manufacturing imperfection on that specific prosthetic foot.

There are some rooms for improvement in the manuscript:

1) Authors have assigned a constant value to the stiffness of each prosthetic foot, based on the linear regression models fit to the force-displacement data, and by which they want to make it easy for clinicians to compare and prescribe prosthetic feet. But they also show that all forefeet and heels exhibited nonlinear mechanical behavior during loading. Defining another value to quantify the nonlinearity is valuable which is done by the authors. But since this nonlinearity is so huge, naming the assigned constant value of the stiffness, linear stiffness is not correct and can mislead the reader.

I would recommend since this numerical value is going to use as an indicator of overall stiffness, it can be called assigned stiffness or something like this which doesn’t make the impression that stiffness values have a linear behavior.

2) I believe it would be beneficial to show both loading and unloading portions of the force-displacement curves in the supporting information. These data can be used for example to study the energy storage and return of prosthetic foot.

3) Some data can still add to Fig 4. to give a better understanding to readers. Regression curve can be shown to illustrate the difference between assigned stiffness and real force-displacement. Highlighting the high and low-loads region of the force displacement can give a quick perception of the defined relative difference between low-load stiffness to high-load stiffness.

4) In Discussion section in line number 278, it is mentioned that data of the current study were compared with data from three previous studies. Sharing this comparison data in a table format would be beneficial for the readers.

5) Since in the Discussion section, the advantages and disadvantages of the current method of mechanical testing are described and it is mentioned that it is comparable with previous methods, it would be better to mention and describe previous methods in the Introduction section.

6. PLOS authors have the option to publish the peer review history of their article (what does this mean?). If published, this will include your full peer review and any attached files.

Reviewer #1: No

Reviewer #2: No

Reviewer #3: **Yes: **Farshid Jalalimoghadas

---

## [Author Response · Author response to Decision Letter 0]

10 Oct 2021

We thank the reviewers for their helpful comments and questions. Our responses to each of the Editor’s, Reviewer 1’s, Reviewer 2’s, and Reviewer 3’s comments are shown below (Editor and Reviewer comments are pasted in italics, each followed by our response). We have made changes and additions to the manuscript accordingly, and believe that it is now stronger. 

Manuscript Number: PONE-D-21-22884

Title: The Effect of Prosthetic Foot Stiffness on Foot-Ankle Biomechanics and Relative Foot Stiffness Perception in People with Transtibial Amputation

Editor:

Response: We have looked through the PLOS One submission requirements as well as the style templates provided, and identified the following changes which we made to ensure consistency with the style template: title page re-formatting, changing figures to the appropriate dimensions, and formatting references. 

 Thank you for stating the following in the Financial Disclosure section: [This research is a project of the Seattle Institute for Biomedical and Clinical Research (https://www.sibcr.org/) supported by the Office of the Assistant Secretary of Defense for Health Affairs, through the Orthotics and Prosthetics Outcomes Research Program (https://cdmrp.army.mil/oporp/default) under Award No. W81XWH-16-1-0569 (PI: D.C.M.). Opinions, interpretations, conclusions and recommendations are those of the author and are not necessarily endorsed by the Department of Defense. The U.S. Army Medical Research Acquisition Activity, 820 Chandler Street, Fort Detrick MD 21702-5014 is the awarding and administering acquisition office. This material is the result of work supported with resources and the use of facilities at the VA Puget Sound Health Care System, Seattle, Washington (https://www.pugetsound.va.gov/). The contents do not represent the views of the United States Department of Veterans Affairs or the United States Government.

This study was not funded by commercial companies and no authors received funding from commercial companies (except that co-authors J.M.C. and C.S.C. are employed by Human Motion Technologies LLC d/b/a Humotech). Humotech was not a funder for this study, and the funder (Department of Defense) had no role in study design, data collection and analysis, decision to publish, or preparation of the manuscript.] 

We note that one or more of the authors are employed by a commercial company: Human Motion Technologies LLC d/b/a Humotech).

Response: It is correct that two of the authors are employed by a commercial company, however, the company was not a funder for this study and the company has no competing interests for the data presented in this manuscript. We updated the wording on the Financial Disclosure Statement to make this clearer, and also included this information in the Competing Interests Statement that we included in the updated Cover Letter. As per the Editor’s comment, we have also looked at the Author Contributions section of the online submission form and have ensured we checked the appropriate boxes for these author’s contributions. However, we do not see any place for free text beyond these checkboxes in the Author Contributions section of the online submission form. Please advise if there is anything further that would be helpful for us to clarify along these lines.

b)Please also include the following statement within your amended Funding Statement. 

Response: We have made the appropriate changes to the Financial Disclosure Statement in response to the Editor’s comment. The added section now reads, “The study sponsor (the Office of the Assistant Secretary of Defense for Health Affairs, through the Orthotics and Prosthetics Outcomes Research Program under Award No. W81XWH-16-1-0569) provided support in the form of salaries for authors [D.C.M., E.G.H., A.T.T., B.J.H]. However, the funder did not have any role in the study design, data collection and analysis, decision to publish, or preparation of the manuscript. The specific roles of these authors are articulated in the ‘author contributions’ section.” We added the updated Financial Disclosures Statement to the resubmission cover letter.

 Please also provide an updated Competing Interests Statement declaring this commercial affiliation along with any other relevant declarations relating to employment, consultancy, patents, products in development, or marketed products, etc.

Response: We have added a Competing Interests Statement to the Cover Letter that includes the requested information. The statement reads, “Co-authors J.M.C. and C.S.C. are employed by Human Motion Technologies LLC d/b/a Humotech, but Humotech was not a funder for this study and the company otherwise has no competing interests that would be relevant to the work done on this manuscript. Therefore, this does not alter our adherence to PLOS ONE policies on sharing data and materials. No authors on this manuscript have any competing interests.”

Response: Thank you. As detailed in our responses above, we have included an updated Funding Statement and Competing Interests Statement in our resubmission cover letter.

Reviewer #1:

Is the manuscript technically sound, and do the data support the conclusions? 

Yes, the research methods seem sound and in line with ISO testing standards. The only question I have about the methods is the addition of a standard shoe for testing, as this just adds a layer of unnecessary material between the device and the surface. From a clinical standpoint it may make sense, but because this is really a characterization of the devices, I feel that the standard shoe is unnecessary but also isn't a problem.

Response: We agree with the reviewer that shoes introduce additional material which has the potential to impact the determined stiffness values. We elected to use a shoe in the testing procedures to maximize clinical relevance. Characterization of foot behavior with a shoe would be most applicable from a clinical standpoint as it is representative of use by patients with lower limb amputation. We agree with the reviewer that because the shoe was standardized across the prosthetic feet tested, and much of the data presented is done so for the purpose of comparison, the offset in stiffness values created will be consistent across feet and therefore would not pose as an issue.

Has the statistical analysis been performed appropriately and rigorously? Yes, the statistical analysis was clearly described and looks appropriate. Have the authors made all data underlying the findings in their manuscript fully available? Yes, the supplemental files contain graphs of all of the force/displacement curves from all of the trials of both the forefoot unloading and heel loading. Data is only presented in a graph, which is sufficient, but possibly the raw data could be added in a file for others to use, particularly for doing further analysis of non-linear behaviors.

Response: In addition to providing the graphs, we have now added mechanical testing data from the prosthetic feet as a supplementary file (S6 Appendix).

Is the manuscript presented in an intelligible fashion and written in standard English? 

Yes, the English is fluid and highly readable. The phrasing is clear and descriptions understandable. Remarkably I don't have any typos or grammatical changes.

Response: Thank you for taking the time to carefully read through our manuscript. We appreciate the positive feedback.

 This was exactly the information I was recently looking for and found quite lacking in the literature so I am really happy to see someone working on this. The authors have selected a wide range of ESR ankles to test in a wide range of sizes and weight classes to get a general idea of the change in stiffness as these variables changed. This is a really good data set to have available for everyone interested in current ankle design stiffness and the relative differences between sizes, weight and manufacturer. Overall, I think it is a nice study that certainly should be published.

Response: Thank you for this positive comment and affirmation of our work. 

Overall the method is clear and the results are understandable, but there are some outliers in the data that I would hope are not due to the setup. It seems that each device was run once, so variation due to the setup can't really be determined.

Response: Outliers observed in the data are unlikely to be due to the setup; we performed mechanical testing on a subset of these prosthetic feet, repeating testing on separate days and found the mechanical testing data to be consistent across different rounds of testing. These repeated testing data are part of a separate manuscript which is currently under review at a different peer-reviewed journal (therefore we haven’t cited this work since it has not yet been accepted for publication). 

 These are the type of data I wish the manufacturers would publish, and I don't know if the authors have reached out to any of them to see if they would provide numbers before or during the study. Another comment is that spring constants for mass produced steel springs generally don't have very accurate spring constants. An average compression die spring usually has a +/- 20% tolerance, while a 'high precision die spring' tends to have something like +/-10%. Although some carbon springs can be held to 2%. I don't think the manufacturers ever state this kind of value or what the reject values would be for their production, but it is possible that there is large variation even within the same device size and category which this study wouldn't be able to capture.

Response: We agree with the reviewer that there can be variation in spring material properties and behaviors, and thus possible variation across devices of the same size and category might exist. We have also found that these data are not reported by the manufacturers. Therefore, to address this comment, we have added a supplemental section, titled S1 Appendix, that includes data from testing of several duplicate units of the same size and stiffness categories to explore any differences that might exist between units. Ten pairs of duplicate feet were tested and minimal differences in calculated linear stiffness were found within pairs (mean difference of 2.4 ± 1.3%). In this supplemental file we also included force-displacement curves for the testing of duplicate units to demonstrate the similarity within pairs.

 One comment about these studies of linear stiffness estimation of the prosthesis is that it is a bit hard to compare to joint torque data that is generally produced from biomechanical studies. I don't know if there is a way to easily convert from the forces to a joint angle approximation, as it would require a bit more information about the way the prosthesis was deforming and the geometry of the device. Defining exactly where on each of the devices contact was made with respect to the geometry of the device is maybe a bit too much to report here, but suggestions would be welcome.

Response: We agree that linear stiffness data does not provide the full picture of prosthetic foot properties and that it is difficult to compare to joint torque data. However, we chose to calculate and report linear stiffness, following the methods described in previous studies for the testing and classification of prosthetic feet by foot stiffness category. Using this method allowed for the comparison of foot stiffness data to that from previous literature (as we discuss in the Discussion section of the manuscript). 

Unfortunately, there is no easy or consistent way to compare prosthetic foot stiffness data to ankle joint torque data calculated from in vivo biomechanical studies. The quantification of ankle joint angle in prosthetic feet is challenging because most prosthetic feet do not have a fixed center of rotation. However, the Reviewer’s suggestion is a good one and perhaps future experiments could define a reference point about which ankle torque and angle could be estimated depending on each foot’s deformation (e.g., based on the motion of the center of pressure during loading). This unfortunately is not something that can easily be derived from the data collected.

 I wish there was a better figure showing the direction of the forces applied to the prosthesis including the positions the displacement is measured from. This would make it clear that it is -15 and +20 degrees from the vertical. Similar to diagrams shown in ISO 10328:2016.

Response: Based on the reviewer’s suggestion we have added a new figure, Fig 2, that indicates the positioning of the foot relative to the loading platform, and the direction in which the foot was loaded.

 Please convert and report all values in SI, even though I know the references you are using are in pounds and pound force.

Response: We made the appropriate changes to present all values in SI units. In cases in which pounds were previously presented, we now present the value in kilograms first, followed by the corresponding value in pounds to allow for easy interpretation by readers accustomed to Imperial units.

Reviewer #2:

This paper presents the mechanical characterization of a number of prosthetic feet, which I believe will be valuable for researchers, and potentially clinicians as well.

Response: We would like to thank the reviewer for taking the time to read and provide comments on the manuscript. We also feel that data like this is lacking and will be valuable for researchers and clinicians.

Major:

 The statistics are not described clearly, or with adequate precision. Rather than describing something like “the significance of the slope of change of delta y over delta x1,” just describe “the significance of the x1 coefficient,” or “significance of the x1-x2 interaction term.”

Response: We have made the requested changes in our description of the statistics.

“Linear mixed effects regression was used to assess associations between calculated forefoot or heel stiffness (dependent variable) and stiffness category (independent fixed effect) with foot size as a fixed effect covariate, modeled as categorical (S3 Appendix). Foot type and foot type by stiffness category interaction were random effects. Stiffness category across feet was scaled so that a value of 1 represented minimum stiffness category (i.e., rated for 59.0 kg (130 lb.)) and 5 represented maximum category (i.e., rated for 113.4 kg (250 lb.)). Scaling was distributed equally within this range depending on the number of manufacturer stiffness categories for each foot type and size. Likelihood ratio tests were carried out to test the hypothesis of no association between calculated stiffness and stiffness category (i.e., the stiffness category coefficient is equal to zero) to test for variability in this association across foot types (by testing the significance of the foot type by stiffness category interaction term). To address our secondary hypothesis, the influence of foot size on the association between calculated stiffness and stiffness category was examined by testing the significance of stiffness category by size interaction terms. Results are summarized with the stiffness category model coefficient (representing the change in calculated stiffness per 1 unit increase in stiffness category) ± standard error (SE), 95% confidence intervals (CI) and marginal R-squares (R2).”

 While obtaining so many feet to test is admirable, I worry that the total content is relatively low for a “full report.” I do not know if PLOS has a short communication format, but I would recommend this more as a short communication.

Response: We respectfully disagree with this comment. the length of this manuscript is commensurate with similar articles, including all those cited in this manuscript. Furthermore, PLOS One does not have a “short communication” format. This study contains a similar number of feet as those cited, but more importantly, it includes a range of types, sizes, and multiple stiffness categories providing data that is not otherwise available and will allow for researchers and clinicians to make comparisons across size and stiffness category for a particular foot. 

 I strongly recommend reworking each of figures 2 and 3 into single-plot figures, using color to distinguish between foot type. As the effect of foot length is marginal, I suggest averaging stiffness across foot-length, instead of showing individual lines for each foot length. Thus, each figure consists of one plot, which consists of five lines, which span stiffness categories. However, the scaling of the categories also presents problems. The numbers are already rather meaningless by themselves, and the mapping to [1,5] is arbitrary enough that I suggest removing it. For this to be simple for clinicians to use as a reference, it makes sense to avoid any additional mapping (or reading, really) that they need to do to understand which foot is which. Perhaps you could change the x-axis into recommended weight (or “medium impact user mean body weight”), and add small numbers next to the points on the line denoting the manufacturer-specific “category”. This would cleanly describe all the most relevant data: difference in keel/heel stiffness between models and between categories (and actual category numbers). In the supplementary information, you could include data as it’s currently shown in Figs 2 and 3 (with foot length included), but perhaps with non-scaled category on the x-axis.

Response: The comparisons made in this paper focus on differences in foot stiffness across stiffness category and size within a foot type, rather than across foot types. For this reason, averaging across foot length would prevent the intended comparisons across the different size feet. While in some cases the differences are marginal, there are instances where this is not the case. We included these plots in part to demonstrate these inconsistencies. For this reason, reworking all plots into single figures would require 15 lines on each figure, which we feel would make it difficult to discern individual differences in stiffness due to size. 

We do however agree that making comparisons across foot type has important relevance for clinical readership. We have made such comparisons across foot type for users of a particular weight ranges in another manuscript which is currently under review at Prosthetics and Orthotics International. 

In response to the reviewer’s comment regarding scaling of categories as the axis label, we agree that this has the potential to cause some confusion. Therefore, we have replaced the scaled category axis labels with the stiffness categories of the feet presented.

 It’s unclear why loading of the heel and unloading of the keel are used in the analysis. Why is heel loading more important than heel unloading, and vice versa for the keel? It seems preferable to measure both the same way. It may also be beneficial to calculate hysteresis, though at such slow loading and unloading rates that may not be realistic.

Response: Thank you for the comment and question. We used loading of the heel and unloading of the keel in the analysis to reflect the stiffness behavior representative of clinical use. Therefore, stiffness was based on foot loading or unloading during the respective phase of gait (i.e., heel loading during the heel contact phase and forefoot unloading during push-off). We have included this in the manuscript on lines 91-93. While hysteresis would provide interesting data to consider, it would be better accomplished using different methodology and could be a good topic for future study.

Minor:

 Figure 1 should contain new images, shot in improved lighting, with improved resolution, and oriented in intuitive ways.

Response: Unfortunately, the mechanical testing equipment used in this study has been deconstructed and thus we are unable to retake any of the photos. The original photo was of higher resolution, but due to the PLOS One file size restrictions, we had to submit it with a slightly lower resolution. To improve the reader’s understanding and clarity of the setup we have added an additional figure (Fig 2) to show how the loading platform was positioned relative to the foot in both heel and forefoot loading conditions, as well as arrows to indicate the direction in which the loading platform moved and thus, load was applied.

 Citations 4, 5, 7 don’t seem to fit the sentence they are supporting particularly well.

Response: Thank you for pointing this out, we replaced the citations with the intended citations (they previously referred to different articles from the same year and first author). 

 “…and internal knee flexion moment during early stance” –do you mean increased internal knee flexion?

Response: We corrected the wording to state, “decreased heel stiffness (i.e., a softer heel) has been associated with prosthetic-side reduced knee flexion and reduced internal knee extensor moment during early stance,” as this is representative of the findings in Adamczyk et al., which was the referenced source.

 “Decreased prosthetic forefoot stiffness 31 (i.e., a softer forefoot) has demonstrated a mix of potentially beneficial effects (e.g., increased prosthetic-side energy storage and return as well as increased ankle peak push-off power and work” –This is somewhat controversial, as many people don’t believe that increasing prosthesis energy storage and return is always beneficial (specifically, energy storage and return continues to increase as prosthesis stiffness is decreased to unreasonably low values). I suggest removing this argument.

Response: We appreciate this comment and have thus removed the classifications of beneficial and detrimental in this sentence. However, decreasing stiffness can increase the prosthetic device energy storage and return, and whether beneficial or detrimental, we feel that it should therefore be included in the effects of decreasing foot stiffness. 

“Decreased prosthetic forefoot stiffness (i.e., a softer forefoot) has demonstrated a mix of effects (e.g., increased prosthetic-side energy storage and return,[9, 13, 14] increased ankle peak push-off power and work,[10, 15] an association with prosthetic side increased knee extensor and hip musculature compensation, increased intact side ankle musculature demand,[13] and increased intact limb loading[9, 11, 16]).”

 “Further, the majority of these studies have not reported linear stiffness across manufacturer-defined stiffness categories within commercial foot types” --it’s not clear what is important about “linear stiffness” in this context… perhaps remove “linear” here?

Response: As suggested we have removed the word “linear.”

“Further, the majority of these studies have not reported stiffness across manufacturer-defined stiffness categories within commercial foot types.”

 How was “neutral” alignment found/defined?

Response: We have added an explanation of the methods used to obtain neutral alignment in the Procedures section.

“Prosthetic foot alignment was set to neutral in all planes (sagittal, coronal, and transverse). Neutral alignment was established in the transverse plane using a level positioned between two points on the shoe, while neutral in the sagittal and coronal planes was established using screws on the prosthetic foot adapter to align the shoe such that the sole of the foot was parallel to the force plate.”

 “a first-order polynomial fit” –maybe just linear fit?

Response: We have modified the language accordingly in the Procedures section and in the S4 and S5 supplemental appendices descriptions.

“Since a minimum force was maintained throughout testing, the data were linearly extrapolated from 50 N to 0 N using the initial ten force-displacement data points.” 

“This was completed using a linear model fit to the initial ten force-displacement data points to estimate the earliest linear stiffness behavior, thus data shown below 50 N of load has been extrapolated.” 

 “(i.e., the stiffness experienced by an average user during heel strike or forefoot push-off).” –At heel strike, I would actually define the load as being exactly 0, but increasing (the moment the heel begins to strike is different than the peak load experienced during the heel strike phase). Please clarify.

Response: We have changed the wording to clarify meaning. Heel stiffness was measured at a point in the loading curve equivalent to the average, effective body weight of a user. Therefore, we have altered the wording to specify the stiffness experienced by an average user during heel loading, as opposed to during heel strike. 

“This was intended to provide an estimate of linear stiffness at an instantaneous, effective body weight load range (i.e., the stiffness experienced by an average user during heel loading or forefoot unloading).”

 Why is foot size a categorical variable? It is a measurable quantity that is definable across models, and you will lose power by treating it as a categorical variable.

Response: While we appreciate this point, we respectfully disagree. We do not consider foot size to be a continuous variable because it contains a finite number of categories. In contrast, if foot size had infinite number of values between any two of the categories, then it may have been appropriate to consider it a continuous variable and treat it as such in our analysis. 

 The statistical test could be described more precisely. For example, “the mean slope of change (i.e., the relationship between amount of stiffness change per stiffness category)” –could this be simplified to: “the coefficient relating category to stiffness.”

Response: We have clarified the statistical methods. Please see our revised description:

“Linear mixed effects regression was used to assess associations between calculated forefoot or heel stiffness (dependent variable) and stiffness category (independent fixed effect) with foot size as a fixed effect covariate, modeled as categorical. Foot type and foot type by stiffness category interaction were random effects. Stiffness category across feet was scaled so that a value of 1 represented minimum stiffness category (i.e., rated for 59.0 kg (130 lb.)) and 5 represented maximum category (i.e., rated for 113.4 kg (250 lb.)). Scaling was distributed equally within this range depending on the number of manufacturer stiffness categories for each foot type and size. Likelihood ratio tests were carried to test the hypothesis of no association between calculated stiffness and stiffness category (i.e., the stiffness category coefficient is equal to zero) and to test for variability in this association across foot types (by testing the significance of the foot type by stiffness category interaction term). To address our secondary hypothesis, the influence of foot size on the association between calculated stiffness and stiffness category was examined by testing the significance of stiffness category by size interaction terms. Results are summarized with the stiffness category model coefficient (representing the change in calculated stiffness per 1 unit increase in stiffness category) ± standard error (SE), 95% confidence intervals (CI) and marginal R-squares (R2).”

 “Effect modification” –what does this mean?

Response: Effect modification refers to whether an effect (in this case the association between calculated stiffness and stiffness category) varies systematically across another variable (in this case foot size). Effect modification is usually tested by adding interaction terms to the model, in this case testing the stiffness category by foot size interaction terms. Please see our revised statistical methods where we clarify this point.

“To address our secondary hypothesis, the influence of foot size on the association between calculated stiffness and stiffness category was examined by testing the significance of stiffness category by size interaction terms.”

 “slope of change” is not a term I have come across, and seems like an odd mix of phrases.

Response: We appreciate that this phrase may have been confusing. We have removed the phrase and simplified the language accordingly: 

“Likelihood ratio tests were carried out to test the hypothesis of no association between calculated stiffness and stiffness category (i.e., the stiffness category coefficient is equal to zero) to test for variability in this association across foot types (by testing the significance of the foot type by stiffness category interaction term). To address our secondary hypothesis, the influence of foot size on the association between calculated stiffness and stiffness category was examined by testing the significance of stiffness category by size interaction terms. Results are summarized with the stiffness category model coefficient (representing the change in calculated stiffness per 1 unit increase in stiffness category) ± standard error (SE), 95% confidence intervals (CI) and marginal R-squares (R2).”

 I recommend writing out the equation of the linear mixed effects model used, and referencing the parameters by their symbols.

Response: We thank the reviewer for this comment and perspective. This was not included originally as all readers may not have a background in statistics, and we thought that including statistical models and equations might take away from the data and comparisons presented. However, in response to this comment we have added the statistical models used in the analyses as a supplemental file titled S3 Appendix, for readers who might gain a better understanding of our methods by having access to these details. Please see the linear mixed effects models below:

Y_ij=β_0+β_1 X+γ_1 W_1+γ_2 W_2+b_0i+b_1i X+e_ij

Where Yij = Stiffness outcome for the ith foot and jth stiffness category measurement (i=1, …, 5 feet; j=1 up to 5 stiffness category measurements)

X = stiffness category (ranging from 1 to 5)

W1 = dummy-coded variable for foot size 28

W2 = dummy-coded variable for foot size 29

0 = mean stiffness outcome at foot size 27, and stiffness category=0—not an interpretable parameter

1 = mean change in stiffness outcome per one unit increase in stiffness category—the main effect of interest

1, �2 = mean difference in outcome between foot size 28 and foot size 29 vs. foot size 27-not the focus of this analysis

b0i = the random effect for foot type i 

b1i = the random effect for foot type i by stiffness category interaction

eij = residual error

For the secondary hypothesis, the following interaction terms were added to the above model:

δ_1 XW_1+δ_2 XW_2

Where the two � coefficients were tested jointly for significance to determine if there was a significant interaction between stiffness category and foot size

 “although variability in stiffness across foot sizes and across consecutive stiffness categories is observed, depending of the type of foot (p<0.01).” –This is a bit unclear to me, consider rewording.

Response: We have edited the statement to address the reviewer’s comment. 

“Linear stiffness was significantly associated with scaled stiffness category (p=0.001), although depending on foot type, variability in stiffness across foot sizes and across consecutive stiffness categories is observed (p<0.01).”

 It’s important to discuss that the moment arm of the heel spring is much shorter, so that even though it is stiffer linearly, the angular stiffness about an approximate ankle joint, or about an instantaneous center of rotation, will not be the same.

Response: We agree with the reviewer that the moment arm of the heel spring is much shorter than that of the forefoot. We noted this in the Fig 4 caption, which states, “On average, heel stiffness values were greater than forefoot stiffness values, across all foot types, which may be attributed to the shorter lever arm of the heel compared to that of the forefoot.” However, a comparison between heel vs. forefoot stiffness was not one of the aims of this study, and we therefore did not include further discussion of this comparison. 

We also agree that linear stiffness does not map to angular stiffness, and therefore it cannot be assumed that the angular stiffness about an approximate ankle joint would change in the same ways as linear stiffness with a shorter lever arm. However, the focus of this paper is on linear stiffness, and therefore angular stiffness was not discussed. While we agree that this is a topic that warrants further discussion, it is one that would likely include differing methods allowing for the determination of a center of pressure and establishment of an approximate ankle joint, and would require future study. 

 Please discuss why a nonlinearity exists in the force-displacement curves. Is the center of pressure moving backwards? Is it material nonlinearity? Other geometric nonlinearity?

Response: Thank you for these questions. The nonlinearity seen in the force-displacement curves is likely due to several contributing factors. The elastic properties of the prosthetic feet are inherently nonlinear given that they are comprised of multiple materials, including composites and/or elastomers. Additionally, the geometry of the varying prosthetic foot designs likely contributes to the nonlinearity of the data. Lastly, as feet are loaded, the center of pressure typically shifts proximally, effectively loading a shorter cantilever beam and likely adding to the non-linearity of the loading data. We have added the following in the Discussion section: 

“The nonlinearity observed on the force-displacement curves is consistent with observations for similar designs of commercial feet tested in previous studies, and is likely due to the elastic properties of materials (e.g., composites and elastomers), geometries of prosthetic feet, and shifting center of pressure during loading. However, consistency in nonlinearity of force-displacement curves across size and category was expected within a given foot type.”

 Figure 4 doesn’t illustrate much new. It would actually be helpful to show a similar set of plots earlier, for readers to better understand the low and high stiffness portions.

Response: We agree with the reviewer that it would be helpful to include a figure that would better demonstrate how the degree of non-linearity was determined. We therefore added S2 Appendix which includes a figure demonstrating the low and high stiffness calculations. S2 Appendix is now referenced earlier in the manuscript to address the reviewer’s comment. Fig 4 (now, Fig 5) was included to illustrate example inconsistencies that can be seen in the force-displacement curves, rather than to highlight the degree of non-linearity. 

 For figures that have curves indicating some sort of gradation of a continuous variable (for example in the supplementary figures), I recommend replacing the random colors with varying shades or darkness of a single color, so that readers don’t have to consistently reference the legend to see if the trend is as expected.

Response: We appreciate this suggestion and have spent time attempting to make these changes. We found that varying shades of a color or using symbols, especially in cases where there were five figures on a single plot or overlapping curves, was less legible and made interpretation of the plots difficult. We chose the colors that we did because they are accessibility colors that are distinct for most readership, and while they require reference to the legend, will allow for the most clarity in the figures. 

 I recommend referencing the supplementary material more explicitly.

Response: We followed the PLOS One Supporting Information Guidelines, and have referred to supplemental material as suggested. For example:

“Forefoot (S4 Appendix) and heel (S5 Appendix) force-displacement curves demonstrated nonlinear force-displacement behaviors under load.”

 I recommend mentioning manufacturing tolerances as a source of this error, and noting that the testing of individual feet is subject to this variability.

Response: While we are not certain which error the reviewer is referring to, we believe that this comment may be in reference to the absence of data from duplicate models of feet (i.e., feet of the same type, size, and stiffness category). Provided this is the case, we responded to this comment and comments by the other reviewers by including data showing a comparison between duplicate models. We have added a supplemental section, titled S1 Appendix, that includes data from testing of several duplicate units of the same size and stiffness categories to explore any differences that might exist between units. Ten pairs of duplicate feet were tested and minimal differences in calculated linear stiffness were found within pairs (mean difference of 2.4 ± 1.3%). We also included force-displacement curves for the testing of duplicate units to demonstrate the similarity within pairs (Fig A in S1 Appendix).

Reviewer #3:

In this manuscript a number of prosthetic feet from five manufacturers in different sizes and stiffness categories has been studied by the authors to compare the linear stiffness properties of prosthetic forefeet and heels. The study has been conducted scientifically and quite all the questions which come in mind during reviewing this manuscript, has been answered in the context. Although, since all the experiments on a prosthetic foot has been repeated on the same prosthetic foot (loading and unloading for six consecutive cycles on the same foot) and the experiments has not been repeated on another prosthetic foot with the same characteristic (manufacturer, size, stiffness category), some unexpected results, like a decrease in measured stiffness as foot stiffness category increased, could happen because of the manufacturing imperfection on that specific prosthetic foot.

Response: Thank you for these comments, we are happy to hear that most of your questions had already been addressed in the manuscript. We agree that determining variation between feet of the same type, size, and stiffness category is important when discussing sources of inconsistencies in calculated stiffness and in the force-displacement curves. To address this comment, and a comment made by Reviewer 1, we have added a supplemental section, titled S1 Appendix, that includes data from testing of several duplicate units of the same size and stiffness categories to explore any differences that might exist between units. Ten pairs of duplicate feet were tested and minimal differences in calculated linear stiffness were found within pairs (mean difference of 2.4 ± 1.3%). We also included force-displacement curves for the testing of duplicate units to demonstrate the similarity within pairs (Fig A in S1 Appendix).

 Authors have assigned a constant value to the stiffness of each prosthetic foot, based on the linear regression models fit to the force-displacement data, and by which they want to make it easy for clinicians to compare and prescribe prosthetic feet. But they also show that all forefeet and heels exhibited nonlinear mechanical behavior during loading. Defining another value to quantify the nonlinearity is valuable which is done by the authors. But since this nonlinearity is so huge, naming the assigned constant value of the stiffness, linear stiffness is not correct and can mislead the reader.

I would recommend since this numerical value is going to use as an indicator of overall stiffness, it can be called assigned stiffness or something like this which doesn’t make the impression that stiffness values have a linear behavior.

Response: As per the reviewer’s recommendation, we have made the following changes: In the Methods section, under the subsection Linear stiffness calculation, we updated language to clarify the procedure used to establish stiffness values. Additionally, we have updated language in the manuscript to include the word “calculated” anytime we are referring to stiffness that was determined using the procedure laid out in the Methods section. For example:

“Calculated linear stiffness was determined using linear regression models fit to two data points (i.e., the minimum point (50 N) and the mean target user body weight specified by each manufacturer for the respective foot stiffness category) on the force-displacement curve (Table 1).” 

 I believe it would be beneficial to show both loading and unloading portions of the force-displacement curves in the supporting information. These data can be used for example to study the energy storage and return of prosthetic foot.

Response: While including both the unloading and loading portions of the force-displacement curves can provide additional information, such as energy storage and return properties of the foot, those details are outside of the scope of this study. We did not include both portions of the curves on these plots, as doing so would render trends and inconsistencies across sizes and categories less clear (i.e., would create twice as many curves with additional overlap). Further, while feet are generally loaded (heel) and unloaded (keel) in the orientation we tested, they are not unloaded (heel) or loaded (keel) in that same orientation during walking. We’re isolating two parts of the gait cycle. Although our approach is limited in this regard, the portions of the respective data align relatively well with walking conditions. 

 Some data can still add to Fig 4. to give a better understanding to readers. Regression curve can be shown to illustrate the difference between assigned stiffness and real force-displacement. Highlighting the high and low-loads region of the force displacement can give a quick perception of the defined relative difference between low-load stiffness to high-load stiffness.

Response: In response to this comment and a comment made by Reviewer 2, we have added a new figure in a supplemental file titled, S2 Appendix, which is referenced in the Procedures section under the sub-section, “Relative difference between low-load stiffness to high-load stiffness.” 

 In Discussion section in line number 278, it is mentioned that data of the current study were compared with data from three previous studies. Sharing this comparison data in a table format would be beneficial for the readers.

Response: We appreciate this suggestion. We have add the suggested table in the supplemental materials (S7 Appendix) and referred to this table in the pertinent section of the Discussion section. 

 Since in the Discussion section, the advantages and disadvantages of the current method of mechanical testing are described and it is mentioned that it is comparable with previous methods, it would be better to mention and describe previous methods in the Introduction section.

Response: We agree with the reviewer’s suggestion. However, we feel this would fit better in the discussion section where we put our methods in context with the scope of previous work, rather than in the introduction section. We have therefore added the suggested comparison of methods across studies to the manuscript as follows:

“In addition to stiffness quantification, methods of mechanically testing prosthetic feet vary across studies. Although there have been numerous previous studies that have mechanically tested prosthetic feet, we identified three such studies that were similar to the current study in the types of prosthetic feet (size, stiffness category and model i.e., Rush HiPro,[18] AllPro,[18] Vari-Flex,[16] and Seattle Lightfoot2[16, 17]) and pylon progression angles tested. In the current study we used an R2000 rotopod for mechanical testing. Womac et al. used the same mechanical testing equipment,[16] whereas Major et al. used a materials testing machine and sine plate (as described in ISO standard 10328),[17, 25] and Koehler et al. used a materials testing machine with the foot fastened to an aluminum bar.[18] While in the current study we tested feet at -15° and +20° degrees sagittal pylon progression angle, Womac et al. tested at 15 different sagittal pylon progression angles ranging from -15° to +30° (including the two pylon progression angles used in the current study),[16] Major et al. tested at -15° and +20° (the two angles tested in the current study) as well as a level loading surface,[17] and Koehler et al. tested only at a +20° pylon progression angle.[18] Target loading thresholds and the number of loading cycles also varied by study. The current study used the manufacturer-specified maximum allowable user body weight for each foot stiffness category when determining loading thresholds. Six loading cycles were performed per foot-pylon-progression-angle combination, with a 50 N minimum load. Womac et al. used the vertical ground reaction force from previously tested human subjects (people with transtibial amputation) walking in a gait laboratory as pylon-progression-angle-specific target loads and one cycle per foot-pylon-progression-angle combination.[16] Major et al. used a target load of 1230 N with 50 N of preloading and two cycles per foot-pylon-progression-angle combination.[17] Koehler et al. individualized target loads by using the body weight of three individuals with transtibial amputation walking with weighted vests.[18] The differences in calculated linear stiffness values between the current study and these previous studies were generally small (see S7 Appendix which details differences in forefoot and heel calculated stiffness across studies for comparable prosthetic feet and pylon progression angles), suggesting that the current testing methods are comparable with previous methods.”

 

References

1. Versluys R, Beyl P, van Damme M, Desomer A, van Ham R, Lefeber D. Prosthetic feet: state-of-the-art review and the importance of mimicking human ankle-foot biomechanics. Disabil Rehabil Assist Technol. 2009;4(2):65-75. doi: 10.1080/17483100802715092. PubMed PMID: 19253096.

2. Bowker JH, Michael JW. Atlas of limb prosthetics: surgical prosthetic, and rehabilitation principles. 2nd ed. St. Louis, MI: Mosby Year Book; 1992.

3. Hofstad C, Linde H, Limbeek J, Postema K. Prescription of prosthetic ankle-foot mechanisms after lower limb amputation. Cochrane Database Syst Rev. 2004;(1):Cd003978. Epub 2004/02/20. doi: 10.1002/14651858.CD003978.pub2. PubMed PMID: 14974050.

4. Stark G. Perspectives on How and Why Feet are Prescribed. JPO. 2005;17(Supplement):S18-S22.

5. van der Linde H, Hofstad CJ, Geurts AC, Postema K, Geertzen JH, van Limbeek J. A systematic literature review of the effect of different prosthetic components on human functioning with a lower-limb prosthesis. J Rehabil Res Dev. 2004;41(4):555-70. doi: 10.1682/jrrd.2003.06.0102. PubMed PMID: 15558384.

6. Webber CM, Kaufman K. Instantaneous stiffness and hysteresis of dynamic elastic response prosthetic feet. Prosthet Orthot Int. 2017;41(5):463-8. doi: Instantaneous stiffness and hysteresis of dynamic elastic response prosthetic feet. PubMed PMID: 28008788.

7. Hansen AH, Starker F. Prosthetic Foot Principles and Their Influence on Gait. Handbook of Human Motion. Cham: Springer International Publishing; 2018. p. 1343-57.

8. Adamczyk PG, Roland M, Hahn ME. Sensitivity of biomechanical outcomes to independent variations of hindfoot and forefoot stiffness in foot prostheses. Human movement science. 2017;54:154-71. doi: 10.1016/j.humov.2017.04.005. PubMed PMID: 28499159.

9. Zelik KE, Collins SH, Adamczyk PG, Segal AD, Klute GK, Morgenroth DC, et al. Systematic variation of prosthetic foot spring affects center-of-mass mechanics and metabolic cost during walking. IEEE transactions on neural systems and rehabilitation engineering : a publication of the IEEE Engineering in Medicine and Biology Society. 2011;19(4):411-9. PubMed PMID: 21708509; PubMed Central PMCID: PMCPMC4286327.

10. Fey NP, Klute GK, Neptune RR. Optimization of prosthetic foot stiffness to reduce metabolic cost and intact knee loading during below-knee amputee walking: a theoretical study. J Biomech Eng. 2012;134(11):111005. doi: 10.1115/1.4007824. PubMed PMID: 23387787; PubMed Central PMCID: PMCPMC3707817.

11. Major MJ, Twiste M, Kenney LP, Howard D. The effects of prosthetic ankle stiffness on stability of gait in people with transtibial amputation. J Rehabil Res Dev. 2016;53(6):839-52. doi: 10.1682/JRRD.2015.08.0148. PubMed PMID: 28273321.

12. Fey NP, Klute GK, Neptune RR. The influence of energy storage and return foot stiffness on walking mechanics and muscle activity in below-knee amputees. Clin Biomech. 2011;26(10):1025-32. doi: 10.1016/j.clinbiomech.2011.06.007. PubMed PMID: 21777999.

13. Fey NP, Klute GK, Neptune RR. Altering prosthetic foot stiffness influences foot and muscle function during below-knee amputee walking: a modeling and simulation analysis. J Biomech. 2013;46(4):637-44. Epub 2013/01/15. doi: 10.1016/j.jbiomech.2012.11.051. PubMed PMID: 23312827.

14. Halsne EG, Czerniecki JM, Shofer JB, Morgenroth DC. The effect of prosthetic foot stiffness on foot ankle biomechanics and relative foot stiffness perception in people with transtibial amputation. Clin Biomech. 2020;80:105141.

15. Klodd E, Hansen A, Fatone S, Edwards M. Effects of prosthetic foot forefoot flexibility on gait of unilateral transtibial prosthesis users. J Rehabil Res Dev. 2010;47(9):899-910. doi: 10.1682/jrrd.2009.10.0166. PubMed PMID: 21174254.

16. Womac ND, Neptune RR, Klute GK. Stiffness and energy storage characteristics of energy storage and return prosthetic feet. Prosthet Orthot Int. 2019;43(3):266-75. doi: 10.1177/0309364618823127. PubMed PMID: 30688551.

17. Major MJ, Scham J, Orendurff M. The effects of common footwear on stance-phase mechanical properties of the prosthetic foot-shoe system. Prosthet Orthot Int. 2018;42(2):198-207. doi: 10.1177/0309364617706749. PubMed PMID: 28486847.

18. Koehler-McNicholas SR, Nickel EA, Barrons K, Blaharski KE, Dellamano CA, Ray SF, et al. Mechanical and dynamic characterization of prosthetic feet for high activity users during weighted and unweighted walking. PloS one. 2018;13(9):e0202884. doi: 10.1371/journal.pone.0202884. PubMed PMID: 30208040; PubMed Central PMCID: PMCPMC6135372 organizations that could influence their work or pose conflicts of interest.

19. Beck ON, Taboga P, Grabowski AM. Characterizing the Mechanical Properties of Running-Specific Prostheses. PloS one. 2016;11(12):e0168298. doi: 10.1371/journal.pone.0168298. PubMed PMID: 27973573; PubMed Central PMCID: PMCPMC5156386.

20. Geil MD, Parnianpour M, Quesada P, Berme N, Simon S. Comparison of methods for the calculation of energy storage and return in a dynamic elastic response prosthesis. Journal of biomechanics. 2000;33(12):1745-50. Epub 2000/09/28. PubMed PMID: 11006404.

21. Geil MD. Energy Loss and Stiffness Properties of Dynamic Elastic Response Prosthetic Feet. JPO. 2001;13(3):70-3. PubMed PMID: 00008526-200109000-00011.

22. Mason ZD, Pearlman J, Cooper RA, Laferrier JZ. Comparison of prosthetic feet prescribed to active individuals using ISO standards. Prosthet Orthot Int. 2011;35(4):418-24. doi: 10.1177/0309364611421692. PubMed PMID: 22031596.

23. van Jaarsveld HW, Grootenboer HJ, de Vries J, Koopman HF. Stiffness and hysteresis properties of some prosthetic feet. Prosthetics and orthotics international. 1990;14(3):117-24. Epub 1990/12/01. PubMed PMID: 2095529.

24. Klute GK, Berge JS, Segal AD. Heel-region properties of prosthetic feet and shoes. J Rehabil Res Dev. 2004;41(4):535-46. doi: 10.1682/jrrd.2003.02.0025. PubMed PMID: 15558382.

25. Standardization IOf. ISO 10328 Prosthetics—structural testing of lower limb prostheses—

requirements and test methods. ISO 10328:2016(En). International Organization for Standardization (ISO).

26. Standardization IOf. ISO 22675 Prosthetics—testing of ankle-foot devices and foot units—

requirements and test methods. ISO 22675:2016(En). International Organization for Standardization (ISO).

27. Team RC. R: A language and environment for statistical computing. In: Team RC, editor. R Foundation for Statistical Computing. Vienna, Austria2018.

28. AOPA. Prosthetic Foot Project. 2010.

---

## [Decision Letter · Decision Letter 1]

6 Jan 2022

PONE-D-21-22884R1Prosthetic forefoot and heel stiffness across consecutive foot stiffness categories and sizesPLOS ONE

Dear Dr. Morgenroth,

Thank you for submitting your manuscript to PLOS ONE. After careful consideration, we feel that it has merit but does not fully meet PLOS ONE’s publication criteria as it currently stands. Therefore, we invite you to submit a revised version of the manuscript that addresses the points raised during the review process. I appreciate the authors' patience while thy were waiting for the reviewers to provide their feedback. I must commend the authors for the great job to improve the manuscript. One of the reviewers has some comments that is worth addressing before publication. I appreciate your responses to those comments. 

We look forward to receiving your revised manuscript.

Kind regards,

Arezoo Eshraghi, Ph.D.

Academic Editor

PLOS ONE

Reviewers' comments:

Reviewer's Responses to Questions

**Comments to the Author**

1. If the authors have adequately addressed your comments raised in a previous round of review and you feel that this manuscript is now acceptable for publication, you may indicate that here to bypass the “Comments to the Author” section, enter your conflict of interest statement in the “Confidential to Editor” section, and submit your "Accept" recommendation.

Reviewer #1: All comments have been addressed

Reviewer #2: (No Response)

Reviewer #3: All comments have been addressed

2. Is the manuscript technically sound, and do the data support the conclusions?

Reviewer #1: Yes

Reviewer #2: Yes

Reviewer #3: Yes

3. Has the statistical analysis been performed appropriately and rigorously? 

Reviewer #1: Yes

Reviewer #2: I Don't Know

Reviewer #3: Yes

4. Have the authors made all data underlying the findings in their manuscript fully available?

Reviewer #1: Yes

Reviewer #2: Yes

Reviewer #3: Yes

5. Is the manuscript presented in an intelligible fashion and written in standard English?

Reviewer #1: Yes

Reviewer #2: Yes

Reviewer #3: Yes

6. Review Comments to the Author

Reviewer #1: The changes in adding Fig 2 and all of the supplemental data and sections clarify a lot about what was done in the study. Overall I think it is a decent study showing relative differences between these devices and general behavior over a large range of weight class and size.

Reviewer #2: Thank you for the responses. The authors have done a good job responding to the reviewers’ comments, and the paper is improved. I have two remaining points of discussion.

1.

In the review responses, the authors wrote: “We used loading of the heel and unloading of the keel in the analysis to reflect the stiffness behavior representative of clinical use. Therefore, stiffness was based on foot loading or unloading during the respective phase of gait (i.e., heel loading during the heel contact phase and forefoot unloading during push-off).” [and in response to a different question from Reviewer 3] “Further, while feet are generally loaded (heel) and unloaded (keel) in the orientation we tested, they are not unloaded (heel) or loaded (keel) in that same orientation during walking. “

I am not following completely, but it seems that the argument has to do mainly with foot orientation? Though the foot orientation vs load are consistent in the material testing, it’s true they would be different during gait… with heel loading and keel unloading occurring more while the CoP is at the end of their respective cantilevers (I believe). But it’s still not clear to me why heel unloading and keel loading are less important than the heel loading and keel unloading... why aren’t these middle sections as “representative of clinical use?” They occur just as often, store/return similar amounts of energy, and have biomechanical implications… It seems it would be more intuitive to present only the loading phase of each, since we don’t know how much is lost due to hysteresis, making it an awkward comparison between heel and keel stiffnesses.

2.

[Previously, I asked:] Why is foot size a categorical variable? It is a measurable quantity that is definable across models, and you will lose power by treating it as a categorical variable.

[Response:] While we appreciate this point, we respectfully disagree. We do not consider foot size to be a continuous variable because it contains a finite number of categories. In contrast, if foot size had infinite number of values between any two of the categories, then it may have been appropriate to consider it a continuous variable and treat it as such in our analysis.

But the point isn’t that it must be “continuous” (as opposed to “discrete”)… only that it must be “numerical” (as opposed to “categorical.”) It is okay that it can only take a finite number of possible values, such as the few integer foot lengths… it is still numerical—and it has ordering and spacing properties that will add substantial value to the statistical model (and will thus be more correct). Since the statistics are an important part of this paper, I must push on this a bit.

Reviewer #3: Thank you for making this revision. All the questions are answered properly and I'm satisfied with the answers to my comments. The resuts of this paper are valuable for other researchers and cilinitians.

7. PLOS authors have the option to publish the peer review history of their article (what does this mean?). If published, this will include your full peer review and any attached files.

Reviewer #1: No

Reviewer #2: No

Reviewer #3: **Yes: **Farshid Jalalimoghadas

---

## [Author Response · Author response to Decision Letter 1]

18 Mar 2022

We would like to thank the editor and reviewers for their positive comments, and are glad that we have addressed the vast majority of questions and concerns from the previous round of edits. Our responses to the remaining questions from Reviewer 2 are shown below (Reviewer comments are pasted in italics, each followed by our response). 

Manuscript Number: PONE-D-21-22884

Title: The Effect of Prosthetic Foot Stiffness on Foot-Ankle Biomechanics and Relative Foot Stiffness Perception in People with Transtibial Amputation

Reviewer #2:

1. In the review responses, the authors wrote: “We used loading of the heel and unloading of the keel in the analysis to reflect the stiffness behavior representative of clinical use. Therefore, stiffness was based on foot loading or unloading during the respective phase of gait (i.e., heel loading during the heel contact phase and forefoot unloading during push-off).” [and in response to a different question from Reviewer 3] “Further, while feet are generally loaded (heel) and unloaded (keel) in the orientation we tested, they are not unloaded (heel) or loaded (keel) in that same orientation during walking.” 

I am not following completely, but it seems that the argument has to do mainly with foot orientation? Though the foot orientation vs load are consistent in the material testing, it’s true they would be different during gait… with heel loading and keel unloading occurring more while the CoP is at the end of their respective cantilevers (I believe). But it’s still not clear to me why heel unloading and keel loading are less important than the heel loading and keel unloading... why aren’t these middle sections as “representative of clinical use?” They occur just as often, store/return similar amounts of energy, and have biomechanical implications… It seems it would be more intuitive to present only the loading phase of each, since we don’t know how much is lost due to hysteresis, making it an awkward comparison between heel and keel stiffnesses.

Response: 

We appreciate the reviewer giving us an opportunity to further clarify our methods and reasoning. In completing mechanical testing, we chose pylon progression angles (i.e., -15° and +20°) that were consistent with ISO standards 10328 and 22675. These pylon progression angles (i.e., shank angles) correspond to heel loading and forefoot unloading, respectively. These angles were also chosen to enable isolation of the heel and forefoot (keel) components, allowing for the determination of stiffness of each individual component. 

As described in Jacqueline Perry’s seminal textbook, Gait Analysis: Normal and Pathological Function, during walking, initial contact (when prosthetic heel loading occurs) typically occurs between 0% and 2% of the gait cycle, corresponding with a shank angle range of approximately -14° to -16°. Therefore, a pylon progression angle of -15° is consistent with the heel loading phase of gait. The heel rocker ends at approximately 12% of the gait cycle, or at a shank angle of approximately 0°, when the entire foot comes in contact with the ground. Thus, physiologic heel unloading begins after a neutral shank angle (0°) and would not be represented at the -15° angle used in our testing. Similarly, terminal stance occurs between 31% and 50% of the gait cycle, which corresponds with a shank angle between 10° and 30°. During this phase of gait, the forefoot is loaded (as the ankle continues dorsiflexing) and then rapidly unloaded (following peak dorsiflexion). Peak dorsiflexion occurs at approximately 41 to 45% of the gait cycle, which corresponds to approximately 19-21° shank angle. Thus, physiologic forefoot loading would not be represented at the 20° angle used in our testing.

While we agree with the reviewer that stiffness quantification of the “middle sections” (i.e., where heel unloading and forefoot loading occur during physiologic gait) may also be important, in order to capture this information, mechanical testing would need to be completed at different pylon progression angles and would not isolate the forefoot and heel components. We therefore included the following wording in the limitations section to clarify for the reader. “Additionally, quasi-static loading performed at two, sagittal-plane pylon progression angles is not representative of loading throughout the full gait cycle, although it is consistent with previous studies.”

One reason that previous studies have included both loading and unloading data has been to quantify hysteresis as well as energy storage and return. However, neither of these were objectives of the present study.

Lastly, to address the reviewer’s final point, the comparisons of linear stiffness measurements made in the manuscript were within heels and keels, respectively. This study did not aim to compare heel versus forefoot stiffness. 

Reference: 

Perry, Jacquelin, and Judith M. Burnfield. "Gait analysis: Normal and Pathological Function 2nd ed." California: Slack (2010).

2. [Previously, I asked:] Why is foot size a categorical variable? It is a measurable quantity that is definable across models, and you will lose power by treating it as a categorical variable. [Response:] While we appreciate this point, we respectfully disagree. We do not consider foot size to be a continuous variable because it contains a finite number of categories. In contrast, if foot size had infinite number of values between any two of the categories, then it may have been appropriate to consider it a continuous variable and treat it as such in our analysis. But the point isn’t that it must be “continuous” (as opposed to “discrete”)… only that it must be “numerical” (as opposed to “categorical.”) It is okay that it can only take a finite number of possible values, such as the few integer foot lengths… it is still numerical—and it has ordering and spacing properties that will add substantial value to the statistical model (and will thus be more correct). Since the statistics are an important part of this paper, I must push on this a bit.

Response: 

We agree with the reviewer that foot size is a numeric variable (more precisely, an integer variable with ordered levels – much like shoe size). However, this does not preclude modeling foot size as categorical. We justify our decision by presenting our model (i.e., foot size modeled as categorical) compared to the reviewer’s suggested change to the model (i.e., foot size modeled as numerical) of calculated stiffness in a bit more detail and by discussing the implications below. 

Here are the estimated fixed effects when modeling size as numerical, as the reviewer suggested (StfCat is the foot stiffness category variable, Size..cm. is the size variable):

Fixed effects:

 Estimate Std. Error t value

(Intercept) 16.72230 17.26553 0.969

StfCat 3.83384 0.33007 11.615

Size..cm. 0.04186 0.61216 0.068

In contrast, here are the estimated fixed effects used in our model, where foot size is modeled as categorical (constructed using two dummy variables with size 27 as the reference category):

Fixed effects:

 Estimate Std. Error t value

(Intercept) 16.95560 2.02977 8.353

StfCat 3.81464 0.30683 12.432

Size28 2.87000 1.12050 2.561

Size29 0.01737 1.13810 0.015

The numerical size variable estimates a slope of change close to zero (i.e., 0.04). However, when the size variable is modeled as categorical, it shows that size 28 has a mean stiffness 2.9 N/mm greater than that for size 27, and is significant at p < .05 (as shown by the t-value). This also affects the slope term for the stiffness category variable, which is estimated with more precision when size is modeled as categorical (as demonstrated by the higher t-value in the 2nd model). 

Below is a comparison of the goodness of fit statistics for model 1 (foot size as continuous) vs. model 2 (foot size as categorical):

 npar AIC BIC logLik deviance Chisq Df Pr(>Chisq) 

Model 1 5 324.76 334.89 -157.38 314.76 

Model 2 6 318.40 330.55 -153.20 306.40 8.3641 1 0.003827 **

Model 2 demonstrates a lower AIC and BIC, with the likelihood ratio test favoring Model 2 compared to Model 1 (p = .0038). This finding supports our decision to model foot size as categorical. The reason for the improved precision of Model 2 compared to Model 1 using a numerical variable is likely due to the non-linear relationship between foot size and foot stiffness that was observed in these data (i.e., size 28cm had increased stiffness relative to both size 27 and 29cm). 

We did not want to make any a prior assumptions as to the association between foot size and calculated stiffness (e.g., whether a linear relationship existed), and with only three size values (i.e., 27, 28, and 29cm), we could not model a nonlinear association with size as numerical. If we had more foot sizes, then modeling size as a numerical variable would potentially result in an improved model fit by adding polynomial or spline terms. However, given that we included three foot-sizes in this study and the model comparisons we presented above, we believe that the model we used is stronger.

---

## [Editor Report · Decision Letter 2]

25 Apr 2022

Prosthetic forefoot and heel stiffness across consecutive foot stiffness categories and sizes

PONE-D-21-22884R2

Dear Dr. Morgenroth,

We’re pleased to inform you that your manuscript has been judged scientifically suitable for publication and will be formally accepted for publication once it meets all outstanding technical requirements.

Kind regards,

Arezoo Eshraghi, Ph.D.

Academic Editor

PLOS ONE
---

## [Editor Report · Acceptance letter]

1 May 2022

PONE-D-21-22884R2 

Prosthetic forefoot and heel stiffness across consecutive foot stiffness categories and sizes 

Dear Dr. Morgenroth:

I'm pleased to inform you that your manuscript has been deemed suitable for publication in PLOS ONE. Congratulations! Your manuscript is now with our production department. 

Kind regards, 

on behalf of

Dr. Arezoo Eshraghi 

Academic Editor

PLOS ONE